# Natural genetic variation in *Arabidopsis thaliana* defense metabolism genes modulates field fitness

Rachel Kerwin[1], Julie Feusier[1,2], Jason Corwin[1], Matthew Rubin[3], Catherine Lin[1], Alise Muok[1,4], Brandon Larson[1,5,6], Baohua Li[1], Bindu Joseph[1], Marta Francisco[1,7], Daniel Copeland[1], Cynthia Weinig[2], Daniel J Kliebenstein[1,8]*

[1]Department of Plant Sciences, University of California, Davis, Davis, United States; [2]Department of Genetics, University of Utah, Salt Lake City, United States; [3]Department of Botany, University of Wyoming, Laramie, United States; [4]Department of Biochemistry, Cornell University, Ithaca, United States; [5]US Department of Agriculture Plant Soil and Nutrition Research Unit, Cornell University, Ithaca, United States; [6]Boyce Thompson Institute for Plant Research Sciences, Faculty of Science, Cornell University, Ithaca, United States; [7]Misión Biológica de Galicia, Pontevedra, Spain; [8]DynaMo Centre of Excellence, Department of Plant and Environmental Sciences, Faculty of Science, University of Copenhagen, Copenhagen, Denmark

**Abstract** Natural populations persist in complex environments, where biotic stressors, such as pathogen and insect communities, fluctuate temporally and spatially. These shifting biotic pressures generate heterogeneous selective forces that can maintain standing natural variation within a species. To directly test if genes containing causal variation for the *Arabidopsis thaliana* defensive compounds, glucosinolates (GSL) control field fitness and are therefore subject to natural selection, we conducted a multi-year field trial using lines that vary in only specific causal genes. Interestingly, we found that variation in these naturally polymorphic GSL genes affected fitness in each of our environments but the pattern fluctuated such that highly fit genotypes in one trial displayed lower fitness in another and that no GSL genotype or genotypes consistently out-performed the others. This was true both across locations and within the same location across years. These results indicate that environmental heterogeneity may contribute to the maintenance of GSL variation observed within *Arabidopsis thaliana*.

*For correspondence: kliebenstein@ucdavis.edu

## Introduction

High levels of standing variation have often been observed among many natural plant and animal populations. This is particularly true for the model species *Arabidopsis thaliana*, which exhibits variation both within and among natural populations and/or accessions (*Pigliucci and Marlow, 2001*; *Atwell et al., 2010*; *Bomblies et al., 2010*; *Chan et al., 2010*; *Platt et al., 2010*; *Cao et al., 2011*; *Debieu et al., 2013*; *Joseph et al., 2013*; *Long et al., 2013*; *Anwer et al., 2014*; *Li et al., 2014*). Models based on mutation-selection balance theory predict that this observed variation will be due to rare alleles at many loci introduced through random mutations that evolution acts on to eliminate through persistent purifying natural selection (*Kimura, 1968*; *Turelli, 1984*). In agreement, studies of nucleotide variation in *Arabidopsis* have found an excess of low frequency polymorphisms relative to expectation (*Purugganan & Suddith, 1998*, *1999*). However, other studies cloning causal genetic variants from natural *Arabidopsis* accessions have found several intriguing examples of intermediate

**eLife digest** 'Genetic variation' describes the naturally occurring differences in DNA sequences that are found among individuals of the same species. These genetic differences arise from random mutations and may be passed on to their offspring. Some of these mutations may improve the ability of an individual to survive and reproduce—known as fitness—and are likely to become more common in the population. Other mutations may reduce an individual's fitness and are likely to be lost. However, it is believed that most of the mutations will have no effect on the fitness of individuals.

It is not known why many of these 'neutral' genetic differences are maintained in populations. Some researchers have proposed that they are kept by chance and that there is no direct advantage to the population of keeping them unless these neutral mutations later become beneficial. However, other researchers think that the genetic variation itself may improve the fitness of the population by allowing it to quickly adapt to changes in the environment.

*Arabidopsis thaliana* is a small plant that lives in many different environments and has high levels of genetic variation in many of its physical traits. One of these traits is the production of molecules called glucosinolates, which help the plants to defend against herbivores and infection by microbes. Previous studies have suggested that variation in the genes that make glucosinolates may improve the fitness of *A. thaliana* populations.

To test this idea, Kerwin et al. carried out a field trial using *A. thaliana* plants that were genetically identical except for some of the genes involved in the production of glucosinolates. Kerwin et al. grew the plants in several different environments over several years. The field trial shows that variation in these genes affected the fitness of the plants in each of the different environments. However, the fitness benefit depended on the environment, and no single gene variant provided the best fitness across all environments, or over all the years of the trial.

Kerwin et al.'s findings suggest that changes in the environment may contribute to the maintenance of genetic variation in the genes that make glucosinolates. This raises the questions of how many other genes in plants (or other species such as humans) have genetic variation that contributes to fitness across varied environments; and how can this link be tested in natural settings.

frequency alleles maintained at polymorphic loci (*Johanson et al., 2000*; *Long et al., 2000*; *Li et al., 2014*). This variation among loci has led to a long-standing interest in elucidating to what extent this genetic variation is neutral in origin or, alternatively, maintained through selective forces (*Levene, 1953*; *Hedrick et al., 1976*; *Bull, 1987*; *Stahl et al., 1999*; *Prasad et al., 2012*).

The neutral theory posits that the majority of genetic polymorphisms have no effect on fitness and that stochastic evolutionary processes, such as genetic drift and migration, are sufficient to explain the genetic and phenotypic variation observed within and among populations (*Darwin, 1859*; *Kimura, 1968*; *Duret, 2008*). This hypothesis has generated numerous modeling studies demonstrating that the standing level of genetic variation in traits can be explained by the demographic history of a species not linked to fitness of an individual (*Wolf et al., 2000*; *Barton and Turelli, 2004*; *Hufford et al., 2012*; *Pyhajarvi et al., 2013*). However, for many ecologically important traits, phenotypic variation has been shown to empirically impact fitness in natural populations, suggesting that natural selection also plays an important role in the evolution of such traits (*Mothershead and Marquis, 2000*; *Adler et al., 2001*; *Tian et al., 2003*; *Korves et al., 2007*; *Milla et al., 2009*). A key step necessary to begin to resolve these discrepancies between theory and empirical observations requires the validation of fitness consequences of variation at specific loci or pathways in the field (*Turelli and Barton, 2004*; *Fournier-Level et al., 2011*; *Hancock et al., 2011*).

Determining the impact of polygenic variation upon fitness in the field informs our understanding of the potential selective and non-selective evolutionary processes that protect or maintain phenotypic variation within a species, such as genetic drift and balancing selection (*Kimura, 1968*; *Hedrick et al., 1976*; *Mitchell-Olds et al., 2007*; *Mojica et al., 2012*). However, most population level studies of evolution and selection in the field have focused on polygenic populations and have been unable to validate the link between variation at specific underlying genes and the resulting fitness consequences of this variation (*Lande and Arnold, 1983*; *Mitchell-Olds and Rutledge, 1986*;

*Gillespie and Turelli, 1989*; *Orr, 1998*). Studies using structured mapping populations, such as *Arabidopsis* RILs, can only associate large genomic regions, rather than individual genes, with quantitative variation in fitness (*Weinig et al., 2003*; *Stinchcombe et al., 2004*; *Juenger et al., 2005*; *Malmberg et al., 2005*). More recently, genome wide studies using *A. thaliana* accessions have been able to associate SNPs to fitness in the field and even predict relative fitness of accessions grown in a common garden (*Fournier-Level et al., 2011*; *Hancock et al., 2011*). However, these associations between loci and fitness need more refining to validate the effect of individual genes. Testing if individual genes impact fitness in the field first requires identifying and cloning the causal genes underlying the phenotypic variation of interest (*Mitchell-Olds, 1995*; *Tian et al., 2003*; *Mitchell-Olds et al., 2007*). Then, these natural alleles need to be recreated as single gene lines, which can require approaches such as chemical mutation (e.g., EMS), generation of transgenic individuals via Agrobacterium-mediated transformation, and/or generation of isogenic lines through successive rounds of backcrossing. Therefore, empirical field testing of individual causative polymorphic genes has only been done rarely, and we do not yet have a good understanding of the extent to which individual genes impact fitness in the field (*Tian et al., 2003*; *Schuman et al., 2012*).

A. thaliana has become a key model system and is extremely suitable for characterizing, cloning and validating genes influencing the fitness consequences of underlying natural variation. This is due, in part, to the ease of transformation as well as the abundance of genomic resources available for this organism, including an extensive library of T-DNA insertion lines and natural accessions (*Alonso et al., 2003*). *Arabidopsis* persists in many different environments and experiences selection from both abiotic pressures, such as temperature and precipitation, and biotic pressures, such as insect and pathogen populations that vary temporally and spatially (*Meyerowitz, 1987*; *Richards et al., 2009*). Potentially to maximize fitness across a broad range of biotia, *Arabidopsis* has evolved high levels of natural variation among accessions for many important phenotypic traits, including the defense compounds, glucosinolates (GSLs) (*Stahl et al., 1999*; *Atwell et al., 2010*; *Chan et al., 2010*). GSLs constitute a diverse set of plant-made defensive metabolites restricted primarily to the Brassicales that are partitioned into three classes, indolic, aliphatic and aromatic, depending on their amino acid precursor. These N and S containing compounds are stored in the vacuoles of plant cells until they are activated through tissue damage, which can occur through insect feeding and pathogen attack. Natural genetic polymorphisms found among a suite of aliphatic GSL genes in *Arabidopsis* are responsible for the majority of GSL diversity observed in the leaf tissue (*Figure 1*). These aliphatic GSL genes encode enzymes, transcription factors and activation co-factors that have been identified, cloned and validated in a laboratory setting (*Table 1*) (*Haughn et al., 1991*; *Li and Quiros, 2003*; *Hansen et al., 2007*, *2008*; *Hirai et al., 2007*; *Li et al., 2008*; *Neal et al., 2010*). Previous studies have uncovered links between GSL variation and ecologically important traits in *Arabidopsis*, such as resistance to insect/pathogen damage, flowering time, and growth, suggesting that GSLs play an important role in determining plant fitness (*Mauricio, 1998*; *Kliebenstein et al., 2002*; *Bidart-Bouzat and Kliebenstein, 2008*; *Hansen et al., 2008*; *Burow et al., 2010*; *Kerwin et al., 2011*; *Züst et al., 2011*). Since the genes responsible for the majority of natural polymorphism in aliphatic GSL have been well characterized in a laboratory setting, the GSL pathway in *Arabidopsis* provides a good system for understanding the impact that individual genes might have on fitness in the field (*Kliebenstein et al., 2001b*; *Halkier and Gershenzon, 2006*; *Hansen et al., 2008*). In this study, we tested the fitness consequences of aliphatic GSL variation in the field by utilizing a collection of lines that vary at specific GSL genes in *Arabidopsis* (Col-0), which recreated observed natural variation in the aliphatic GSL pathway found among accessions (*Table 2*) (*Mauricio, 1998*; *Kliebenstein et al., 2002*; *Bidart-Bouzat and Kliebenstein, 2008*; *Hansen et al., 2008*; *Burow et al., 2010*; *Kerwin et al., 2011*; *Züst et al., 2011*).

## Results

### Synthetic laboratory population mimics natural GSL variation in *Arabidopsis*

The GSL profile of a plant is characterized by the presence and relative abundance of the various GSL structures it produces. Among *Arabidopsis* accessions, GSL profiles show extensive phenotypic variation across the species geographic distribution (*Figure 2*) (*Chan et al., 2010*). While previous studies have linked GSL profile variation to insect resistance, as well as correlated the geographic

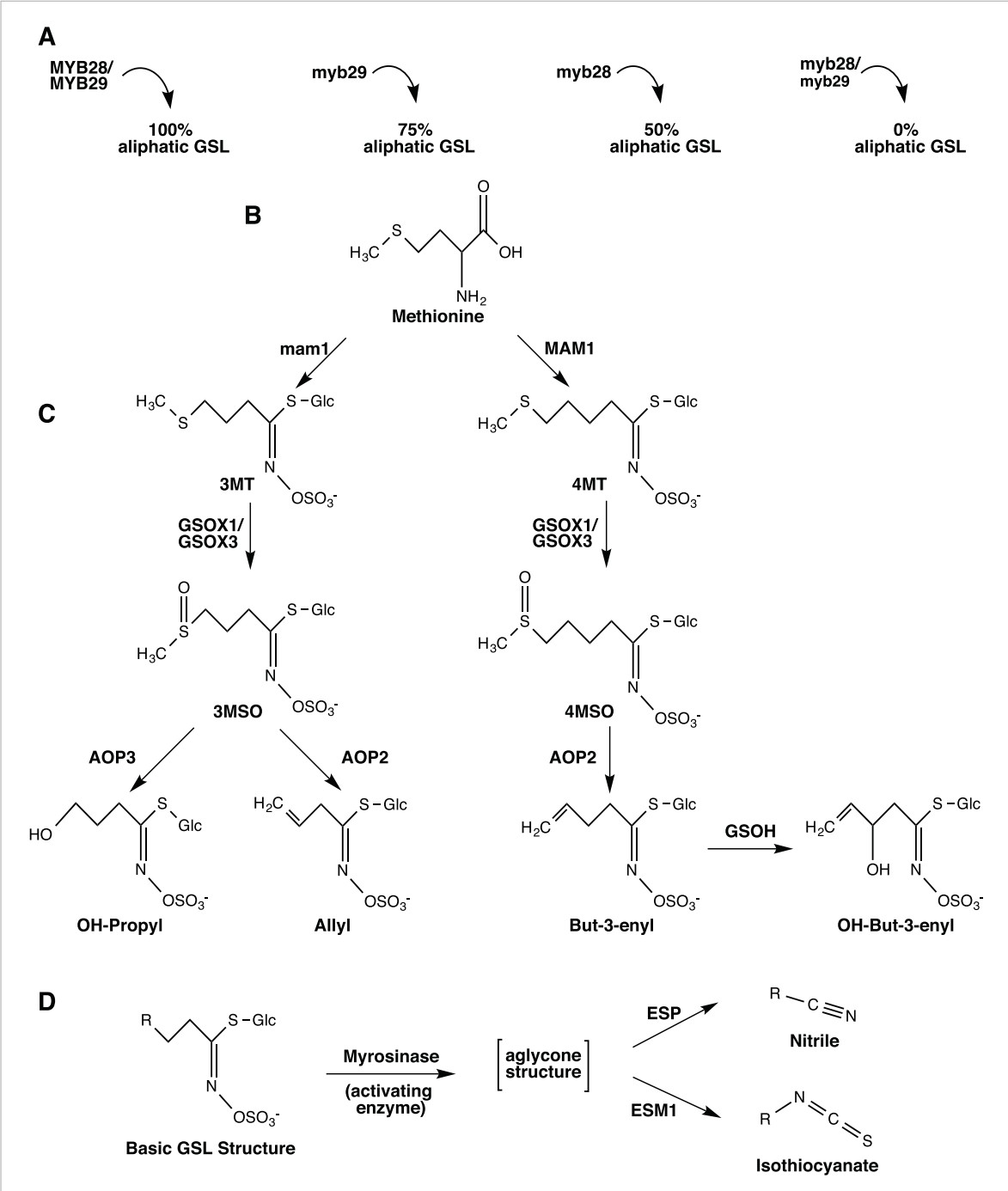

**Figure 1**. Overview of aliphatic GSL biosynthesis and activation in *Arabidopsis thaliana*. Arrows represent the steps involved in aliphatic glucosinolate (GSL) biosynthesis that have been validated through laboratory experiments and are naturally variable within *A. thaliana*. Gene names are listed next to or above the arrows. (**A**) **Regulation of aliphatic GSL biosynthesis**. The transcription factors MYB28 and MYB29 control accumulation of aliphatic GLS. A double knockout in these genes results in no aliphatic GLS accumulation, while a single knockout in these genes leads to a 50% reduction in aliphatic GSL (myb28) or a 25% reduction in aliphatic GSL (myb29), compared to WT Col-0. The biosynthetic enzymes MAM1 and AOP2 also influence aliphatic GLS accumulation and a non-functional allele at either locus leads to decreased GSL accumulation. (**B**) **Amino acid chain elongation**. During chain elongation, carbons are added to a methionine precursor through a series of reactions producing an elongated amino acid. Variation at the Elong locus controls the number of carbons added to the amino acid precursor and therefore, the length of the GSL side chain, R. A functional allele at this locus, MAM1, leads primarily to accumulation of GSL with four carbon (4C) length side chains, whereas a non-functional allele, gsm1 leads to accumulation of GSL with three carbon (3C) length side chains. The elongated amino acid is subsequently converted into a GLS via the core pathway (not shown). (**C**) **Side chain modification**. The GSL compounds produced can then undergo a series of side chain modifications that lead to the suite of diverse GSL compounds

*Figure 1. continued on next page*

*Figure 1. Continued*

found in *Arabidopsis*. Side chain modification is controlled by variation of GSOX1, GSOX3, AOP2, AOP3 and GSOH. GSOX1 & GSOX3 oxygenate a methylthio (MT) to methylsulfinyl (MSO) GSL. AOP2 converts MSO to alkenyl, such as allyl and but-3-enyl. AOP3, on the other hand converts only 3C length MSO to OH-propyl GSL and cannot act on the 4MSO GSL. GSOH oxygenates the 4C but-3-enyl to the OH-alkenyl GSL, OH-but-3-enyl. Since GSOH acts on but-3-enyl GSL, which is a product of AOP2, a functional AOP2 is necessary for GSOH to function and AOP2 is said to be epistatic to GSOH. Col-0 is functional for MAM1 and the GSOX's, null for both AOP2 and AOP3, and functional for GSOH, resulting in accumulation of primarily 4MSO GLS. See *Figure 1—figure supplements 1–17* for images of GSL traces for each GSL genotypes in our mutant laboratory population. (**D**) **GSL Activation**. Once produced, GLS are stored in the vacuole in their stable, unreactive form until activation occurs. Upon cellular disruption, such as occurs during pathogen attack, insect herbivory or even wind damage, GLS come into contact with their own plant-made activating enzyme, myrosinase. After production, myrosinase is stored in vacuoles of idioblastic cells called myrosin bodies. Myrosinase activates the GSL compound by cleaving the glucose moiety, yielding an unstable aglycone structure that non-enzymatically rearranges to either nitriles or isothiocyanates, depending on the presence of the co-activators ESM1 and ESP.

The following figure supplements are available for figure 1:

**Figure supplement 1**. HPLC trace of *Arabidopsis thaliana* accession Columbia-0 wild-type genotype.

**Figure supplement 2**. HPLC trace of *Arabidopsis thaliana* accession Columbia-0 myb28 genotype.

**Figure supplement 3**. HPLC trace of *Arabidopsis thaliana* accession Columbia-0 myb29 genotype.

**Figure supplement 4**. HPLC trace of *Arabidopsis thaliana* accession Columbia-0 gsm1 genotype.

**Figure supplement 5**. HPLC trace of *Arabidopsis thaliana* accession Columbia-0 gsox1 genotype.

**Figure supplement 6**. HPLC trace of *Arabidopsis thaliana* accession Columbia-0 gsox3 genotype.

**Figure supplement 7**. HPLC trace of *Arabidopsis thaliana* accession Columbia-0 AOP2 genotype.

**Figure supplement 8**. HPLC trace of *Arabidopsis thaliana* accession Columbia-0 ESP genotype.

**Figure supplement 9**. HPLC trace of *Arabidopsis thaliana* accession Columbia-0 gsoh genotype.

**Figure supplement 10**. HPLC trace of *Arabidopsis thaliana* accession Columbia-0 myb28/myb29 genotype.

**Figure supplement 11**. HPLC trace of *Arabidopsis thaliana* accession Columbia-0 myb28/gsm1 genotype.

**Figure supplement 12**. HPLC trace of *Arabidopsis thaliana* accession Columbia-0 myb29/gsm1 genotype.

**Figure supplement 13**. HPLC trace of *Arabidopsis thaliana* accession Columbia-0 myb28/AOP2 genotype.

**Figure supplement 14**. HPLC trace of *Arabidopsis thaliana* accession Columbia-0 myb28/gsoh genotype.

**Figure supplement 15**. HPLC trace of *Arabidopsis thaliana* accession Columbia-0 myb29/AOP2/gsoh genotype.

**Figure supplement 16**. HPLC trace of *Arabidopsis thaliana* accession Columbia-0 AOP2/gsoh genotype.

**Figure supplement 17**. HPLC trace of *Arabidopsis thaliana* accession Columbia-0 myb28/myb29/gsoh genotype.

distribution of insect populations with GSL profile-type across Europe, it is still not known to what extent, if at all, individual GSL genes affect fitness in the field (*Mauricio, 1998*; *Bidart-Bouzat and Kliebenstein, 2008*; *Züst et al., 2012*). To test if standing genetic variation within the aliphatic GSL defensive pathway of *A. thaliana* impacts fitness in the field, we utilized an existing set of genotypes that recreate natural variation at eight specific GSL loci, with the reference accession, Col-0, as the genetic background. These transgenic lines consisted of loss-of-function T-DNA insertion lines, an

**Table 1**. Polymorphic genes involved in aliphatic GSL synthesis and activation

| Gene name | Locus | ATG # | Gene type | Gene function | Mutation type in Col-0 |
|---|---|---|---|---|---|
| MYB28 | MYB28 | At5g61420 | TF | Positive regulator of aliphatic GSL (*Sønderby et al., 2007, 2010*) | T-DNA |
| MYB29 | MYB29 | At5g07690 | TF | Positive regulator of aliphatic GSL (*Sønderby et al., 2007, 2010*) | T-DNA |
| MAM1 | MAM1 | At5g23010 | Enzyme | Controls 3C–4C chain elongation (*Haughn et al., 1991; de Quiros et al., 2000; Kroymann et al., 2003*) | EMS |
| GSOX1 | GSOX | At1g65860 | Enzyme | Converts MT to MSO GSL (*Hansen et al., 2007; Li et al., 2008*) | T-DNA |
| GSOX3 | GSOX | At1g62560 | Enzyme | Converts MT to MSO GSL (*Hansen et al., 2007; Li et al., 2008*) | T-DNA |
| AOP2 | AOP | At4g03060 | Enzyme | Converts MSO to alkenyl GSL (*Kliebenstein et al., 2001c*) | 35S OX |
| AOP3 | AOP | At4g03050 | Enzyme | Converts 3MSO to hydroxy-propyl GSL (*Kliebenstein et al., 2001c*) | n/a |
| GSOH | GSOH | At2g25450 | Enzyme | Converts butenyl to OHB (*Hansen et al., 2008*) | T-DNA |
| ESP | ESP | At1g54040 | Co-factor | Guides formation of activated GSL to nitriles (*Lambrix et al., 2001*) | 35S OX |

Shown are the identities, functions and mutation types of nine genes representing seven loci important for aliphatic GSL synthesis and activation. These genes were chosen for mutant laboratory population of *Arabidopsis thaliana* accession Col-0 due to the fact that they represent the majority of aliphatic GSL variation observed in *Arabidopsis*. Each of these genes is naturally polymorphic among *Arabidopsis* accessions.

EMS mutant and gain-of-function overexpression lines that were originally created to validate individual genes as causal for GSL natural variation (*Table 1*). For example, the *AOP2* gene was found to encode an enzyme that converts methylsulfinyl (MSO) GSL into alkenyl GSL (*Figure 1* and *Figure 1—figure supplement 7*) (*Kliebenstein et al., 2001c*). Importantly, the *AOP2* gene is polymorphic among *Arabidopsis* accessions, with Col-0 accession containing a natural knockout that abolishes its function. Therefore, introducing the functional allele back into Col-0 created a single gene mimic of the natural variation found in *Arabidopsis* (*Figure 1* and *Table 1*) (*Kliebenstein et al., 2001c*). The natural variation at the other causal genes has been similarly mimicked as described in the listed citations (*Table 1*). This was facilitated by the fact that all of these genes contain natural presence/absence polymorphisms (citations in *Table 1*).

Each of these transgenic lines had been backcrossed to Col-0 several times to remove unlinked polymorphisms in the original studies (*Table 1*). For this study, the transgenic lines were manually crossed to each other to represent the phenotypic variation in GSL profiles found among *Arabidopsis* accessions (*Table 2*, *Figures 1, 2*). This synthetic laboratory population varies at specific genes controlling aliphatic GSL variation within a single common genetic background. Utilizing this synthetic laboratory population, we can explicitly measure the impact of variation in a suite of aliphatic GSL genes on fitness components in the field without confounding variation in other regions of the genome.

We tested our population in multiple environments, which allowed us to separate the effects of genotype from environment, to determine if traits measured in the field are environmentally controlled. This could be particularly important if selection pressures fluctuate across environments. We transplanted 2 week old, greenhouse-germinated replicates of the synthetic laboratory population into the field at the University of California, Davis in Davis, CA in Spring 2012 and the University of Wyoming in Laramie, WY in Summer 2011 and Summer 2012. In each of our three field trials, which represent three environments, genotypes were replicated in 40 randomized blocks in the field, for a total of 120 blocks/replicates. To distinguish the effects of GSL variation alone from the interaction of GSL variation with field herbivory as well as assess the effects of leaf damage in the field, half of the blocks in each field trial were treated with pesticides and the other half were not (*Figure 3*) (*Mauricio, 1998*).

**Table 2**. Allelic variation of polymorphic aliphatic GSL loci in structured population

| Genotype | MYB28 | MYB29 | MAM1 | GSOX1 | GSOX3 | AOP2 | GSOH | ESP |
|---|---|---|---|---|---|---|---|---|
| Col-0 | + | + | + | + | + | − | + | − |
| myb28 | − | + | + | + | + | − | + | − |
| myb29 | + | − | + | + | + | − | + | − |
| gsm1 | + | + | − | + | + | − | + | − |
| gsox1 | + | + | + | − | + | − | + | − |
| gsox3 | + | + | + | + | − | − | + | − |
| AOP2 | + | + | + | + | + | + | − | − |
| AOP2/gsoh | + | + | + | + | + | + | − | − |
| Gsoh | + | + | + | + | + | − | − | − |
| Myb28/gsoh | − | + | + | + | + | − | + | − |
| Myb28/gsm1 | − | + | − | + | + | − | + | − |
| Myb28/AOP2 | − | + | + | + | + | + | + | − |
| Myb29/gsm1 | + | − | − | + | + | − | + | − |
| Myb29/AOP2/gsoh | + | + | + | + | + | + | − | − |
| Myb28/myb29 | − | − | + | + | + | − | + | − |
| Myb28/myb29/gsoh | − | − | + | + | + | − | − | − |
| ESP | + | + | + | + | + | − | + | + |

Shown are the genotypes in the mutant laboratory GSL population used in this study and the allelic state of each gene within each of them. Each gene in our population is naturally polymorphic among *Arabidopsis* accessions. See **Table 1** for gene functions. For each gene listed, a '+' indicates a functional allele and a '−' indicates a non-functional allele. The loss-of-function and gain-of-function mutant lines shown in **Table 1** were manually crossed to generate this population of genotypes, each of which vary from Col-0 at only these eight genes, including single, double, and triple mutants.

## GSL genetic variation controls GSL profile in the field

Since the genes underlying variation in the aliphatic GSL pathway investigated in this study have been previously validated using lab techniques, we have a solid working knowledge of the resulting laboratory GSL profiles (*Beekwilder et al., 2008*; *Hansen et al., 2008*) (*Figure 1—figure supplements 1–17*). However, these GSL genotypes have not previously been tested in the field to determine if they produce the same GSL profiles as when grown in the laboratory. We particularly wanted to assess if variation at individual aliphatic GSL genes has the same impact on GSL profile in the field as predicted from published lab experiments when the plants are grown in different complex environments, and therefore measured GSL on all the plants grown in each of our three field trials. A mixed model analysis of field GSL revealed that the majority of variation in GSL profiles in the field was controlled by the GSL genotypes that we generated (*Table 3*). Importantly, the majority of the GSL genotypes produced the expected GSL profiles in the field, consistent with the lab studies (*Figure 4* and *Figure 1—figure supplements 1–17*). To quantify the similarity in profiles between field and lab grown samples, we conducted a PCA analysis using the GSL profiles of these genotypes grown in a growth chamber. The first four vectors from our PCA were able to explain >99% of the variation in GSL profile. We utilized the loadings from the chamber PCA to estimate PCA scores of the first four vectors using the chamber GSL and field GSL. The scores for the field grown genotypes were highly correlated with the lab grown genotypes, showing that the GSL genetic variation leads to highly similar field and lab profiles (*Table 4*).

In addition to the quantitative comparison of profiles, we also investigated the specificity of each locus in producing particular GSL structures to ensure that its field behavior mimicked the lab behavior. We found that, for the most part, each GSL gene produced the expected GSL phenotype in the field. For example, all lines harboring a functional *AOP2* gene produce alkenyl GSL (e.g., but-3-enyl GSL) (*Figures 1, 4*). Additionally, the functional/non-functional allelic state at the *MAM1* locus was

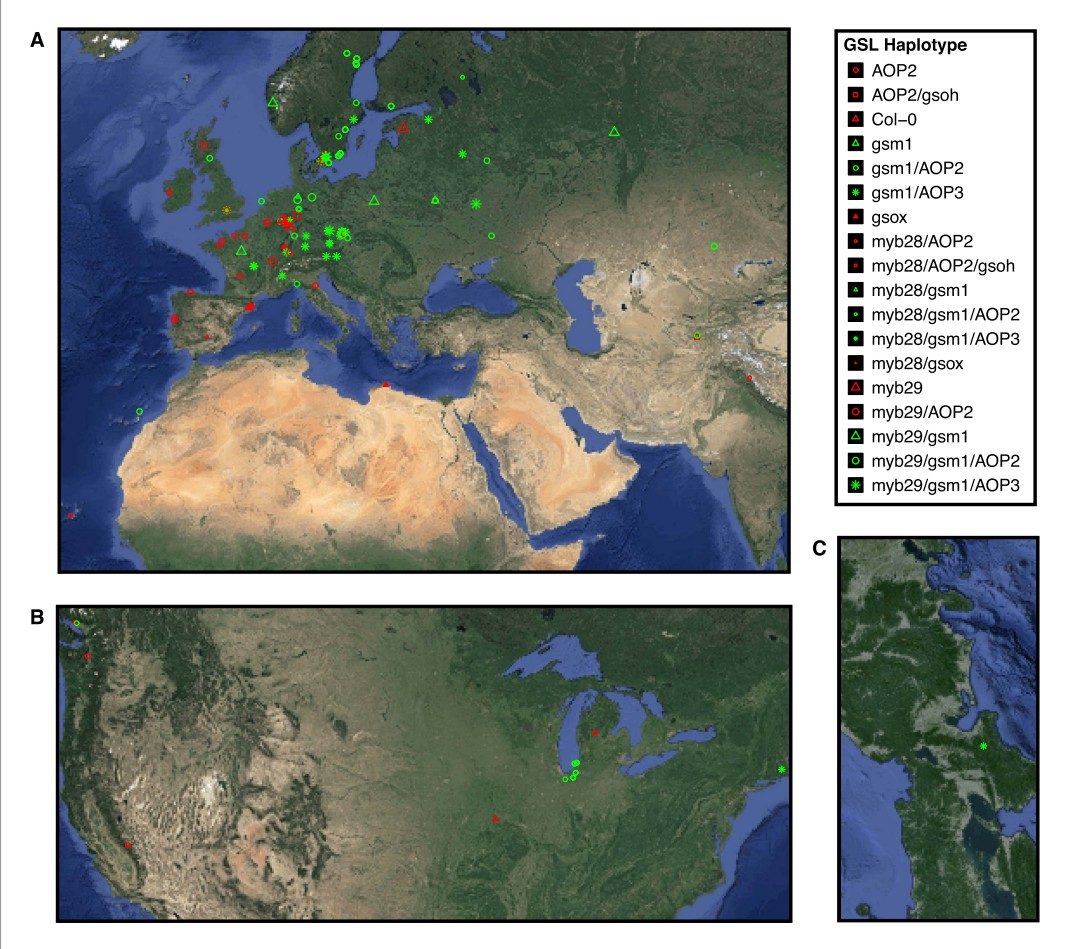

**Figure 2**. Globally distributed collection of *Arabidopsis thaliana* accessions that vary with respect to GSL haplotype. Shown are the geographic origins of 144 *Arabidopsis* accessions across (**A**) Europe and Northern Africa, (**B**) North America and (**C**) Japan, as well as their corresponding GSL haplotypes and chemotypes. GSL haplotype names correspond to allelic identity at six polymorphic loci involved in aliphatic GSL production, based on GSL profile data collected from each accession. Haplotype names use Col-0 as a reference, which is functional at four or the six loci. Symbol shape, color and size indicate GSL chemotype (i.e., phenotype based on GSL profile). Red = 3C (non-functional MAM1), green = 4C GSL (functional MAM1), triangle = MSO (non-functional AOP), square = alkenyl (functional AOP2), circle = OH-alkenyl (functional GSOH), star = OH-Propyl (functional AOP3), point size 1 = 100% accumulation of aliphatic GSL (compared to Col-0), point size 0.5 = 50% accumulation of aliphatic GSL (non-functional MYB28) and point size 1.5 = 75% accumulation of aliphatic GSL (non-functional MYB29). See *Figure 2—source data 1* for table of accession geographic information, *Figure 1* for schematic of biosynthetic pathway and *Figure 9* for more details on the allelic state at each locus for all 18 GSL haplotypes.

The following source data is available for figure 2:

**Source data 1**. Geographic origin and GSL haplotype information for a collection of 144 Arabidopsis thaliana accessions.

always predictive of the chain-length of the GSL in the field as predicted from lab experiments. The lines with a functional *MAM1*, like Col-0, produced more 4C GSL than 3C GSL, while genotypes with a non-functional *MAM1* always produced more 3C GSL than 4C GSL (*Figure 4*) (*Haughn et al., 1991*). A functional copy of *GSOH*, the gene encoding the enzyme to create 2-OH-but-3-enyl, always leads to the production of 2-OH-but-3-enyl GSL from but-3-enyl GSL (*Figures 1, 4*) (*Hansen et al., 2008*). In addition to the biosynthetic genes, the *MYB* genes, which encode transcription factors that control accumulation of aliphatic GSLs, showed similar field phenotypes as were found in the lab (*Hirai et al., 2007*; *Sønderby et al., 2007*, *2010*). Specifically, a non-functional *MYB28* leads to an almost

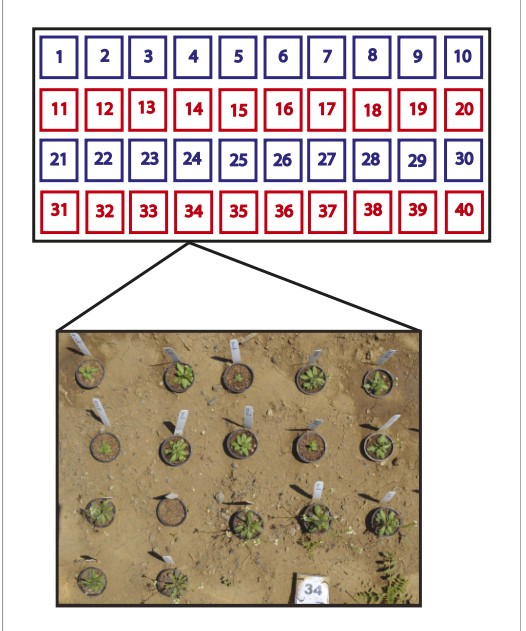

**Figure 3**. Split-plot field trial setup. Shown is the field trial setup used in all three environments. In each environment, 40 blocks were arranged into rows of 10 blocks and each row was called a plot. Within each block, the complete set of 17 genotypes was randomly organized, for a total of 40 genotype replicates per environment. Each plot (four per environment) was placed into one of two treatment groups. The '− Herbivory' treatment group received pesticide application to prevent leaf damage (shown in blue). The '+ Herbivory' or control treatment group did not receive pesticide application (shown in red). This setup was repeated in each of the three environments, for a total of 120 blocks/genotype replicates and 12 plots, split between the two treatment groups. Environment and treatment were nested within plot, making this field trial setup a split-plot design. Seedlings were transplanted from the greenhouse into the field at 2 weeks of age where they were allowed to flower and then subsequently harvested for further analysis in the laboratory.

complete reduction in long chain (8C) GSL and a 60% reduction in short chain (3C and 4C) GSL (*Figure 4*) (*Hirai et al., 2007*; *Sønderby et al., 2007, 2010*). A non-functional *MYB29* leads to a 40% reduction in short chain GSL with no significant reduction in long chain GSL (*Figure 4*) (*Hirai et al., 2007*; *Sønderby et al., 2007, 2010*). A double mutant in *MYB28* and *MYB29* lead to an almost complete loss of all aliphatic GSLs, as expected (*Figure 4*) (*Hirai et al., 2007*; *Sønderby et al., 2007, 2010*). The only genes for which the field and laboratory GSL profile data differ are *GSOX1* and *GSOX3*, which are two tightly linked genes at the GSOX locus that also contains two additional genes, *GSOX2* and *GSOX4*. In the lab, *gsox1* and *gsox3* mutants accumulate higher levels of methylthio (MT) GSL than does Col-0, due to reduced expression of a flavin-monooxygenase that converts the MT to MSO GSL (*Figure 1*) (*Hansen et al., 2007*; *Li et al., 2008*). In the field there was no measureable accumulation of MT GSL in any line, likely due to the redundant function of the *GSOX2* and *GSOX4* genes (*Kerwin et al., 2011, 2012*; *Li et al., 2008*). Thus, the field results show that the laboratory work on GSL genotypes and their associated GSL profiles are translatable and predictive of the GSL profiles found in naturally fluctuating environments.

## Environment and genetic variation interact to control GSL accumulation in the field

Conducting field trials in multiple environments enabled us to test the effect of different environmental conditions on our field traits. The specific composition of GLSs within a genotype largely did not change across the environments (*Table 4*). In contrast, the total amount of aliphatic GSL content, that is, the sum of all aliphatic GSLs per sample, showed a significant genotype by environment effect, indicating that impact of environment on total aliphatic GSL accumulation varied among the different GSL genotypes in this study (*Table 3* and *Figure 5*). For example, the *AOP2* genotype showed a dramatic variation in total aliphatic GSL across the three field trials (*Figure 5*). In contrast, a number of other genotypes tended to show similar accumulation across the environments. For example, genotypes with a *myb28/myb29* double knockout accumulated virtually no GSL in all three environments. Thus, the GSL genotype is the dominant determinant of GSL profile in the field while total aliphatic GSL accumulation is determined by an interaction of genotype and environment within our laboratory population.

## Leaf damage in the field varies across environment

A critical way in which plant environments fluctuate is with respect to insect populations that vary both temporally and spatially in a manner that could have a profound impact on variation in plant damage (*Mauricio, 1998*; *Richards et al., 2009*). To assess if changes in environment impact herbivory levels, we measured leaf damage on a scale from 0–10 in all three field trials, with and without a pesticide

**Table 3.** Mixed model table of leaf damage, flowering time and GSL in the field

| Fixed effects | Leaf damage | | | | | Flowering time | | | | |
|---|---|---|---|---|---|---|---|---|---|---|
| Source of variation | df | SS | MS | F | p value | df | SS | MS | F | p value |
| Genotype | 16 | 280 | 18 | 3.7 | 7.6E-07 | 16 | 7471 | 467 | 9.4 | 5.8E-23 |
| Environment | 2 | 207 | 104 | 7.5 | 0.02 | 2 | 36603 | 18302 | 464.6 | 8.6E-08 |
| Treatment | 1 | 17 | 17 | 0.3 | – | 1 | 107 | 107 | 7.3 | 0.04 |
| Geno:Env | 32 | 617 | 19 | 4.0 | 7.2E-13 | 32 | 2678 | 84 | 3.0 | 5.3E-08 |
| Geno:Trt | 16 | 75 | 5 | 1.0 | – | 16 | 655 | 41 | 1.3 | – |
| Env:Trt | 2 | 32 | 16 | 1.9 | – | 2 | 526 | 263 | 6.5 | 0.03 |
| Geno:Env:Trt | 32 | 201 | 6 | 1.3 | – | 32 | 1115 | 35 | 1.3 | – |

| Random effects | Leaf damage | | | | | Flowering time | | | | |
|---|---|---|---|---|---|---|---|---|---|---|
| Source of variation | df | SS | MS | $\chi^2$ | p value | df | SS | MS | $\chi^2$ | p value |
| Plot(Trt:Env) | 1 | 0 | 0 | 8.7 | 0.003 | 1 | 0 | 0 | 0.0 | – |
| Residual | 1904 | 5 | 0 | NA | – | 1750 | 26 | 0 | NA | – |

| Fixed effects | Total aliphatic GSL | | | | | Total indole GSL | | | | |
|---|---|---|---|---|---|---|---|---|---|---|
| Source of variation | df | SS | MS | F | p value | df | SS | MS | F | p value |
| Genotype | 16 | 2509273 | 156830 | 34.6 | 3.0E-95 | 16 | 33432 | 2090 | 7.3 | 1.1E-16 |
| Environment | 2 | 14588 | 7294 | 1.6 | – | 2 | 1520 | 760 | 0.6 | – |
| Treatment | 1 | 1430 | 1430 | 0.3 | – | 1 | 2208 | 2208 | 4.3 | – |
| Geno:Env | 32 | 305993 | 9562 | 2.1 | 3.3E-04 | 32 | 14139 | 442 | 1.5 | 0.03 |
| Geno:Trt | 16 | 100139 | 6259 | 1.4 | – | 16 | 7488 | 468 | 1.6 | – |
| Env:Trt | 2 | 3938 | 1969 | 0.4 | – | 2 | 610 | 305 | 1.0 | – |
| Geno:Env:Trt | 32 | 116269 | 3633 | 0.8 | – | 32 | 9531 | 298 | 1.0 | – |

| Random effects | Total aliphatic GSL | | | | | Total indole GSL | | | | |
|---|---|---|---|---|---|---|---|---|---|---|
| Source of variation | df | SS | MS | $\chi^2$ | p value | df | SS | MS | $\chi^2$ | p value |
| Plot(Trt:Env) | 1 | 193 | 193 | 72.6 | 5.6E-16 | 1 | 15 | 15 | 67.5 | 2.2E-16 |
| Residual | 1490 | 4342 | 2 | NA | – | 1491 | 269 | 0 | NA | – |

A linear mixed model was fitted to phenotypes measured on plants grown in the field. Variation was partitioned among the fixed effects, Genotype, Environment, and Treatment as well as a random factor, Plot, inside which Treatment and Environment were nested. Phenotypic data used in the model was collected on 17 genotypes from three environments and two treatment groups. df = degrees of freedom, SS = Type II Sums of Squares variation, MS = Mean Squared variation, F = F statistic (for fixed factors), $\chi^2$ = chi squared statistic (for random factors). p value = probability value from either an F test or a chi squared test, depeding on the source of variation. Non-significant p values (>0.05) are represented by a dash.

**Source data 1.** Mixed model table of phenotypes measured in the field.

**Source data 2.** LSMeans of phenotypes measured in the field.

treatment (*Figure 6*). A mixed model analysis showed that leaf damage significantly varied across the three environments but that the pesticide application did not significantly alter leaf damage in the field (*Table 3*). The UWY2012 field trial (mean = 2.610) had significantly higher levels of leaf damage than both UWY2011 (mean = 1.17, p value <1e-04) and UCD2012 (mean = 1.50, p value <1e-04), though UCD2012 and UWY2011 environments did not differ significantly for leaf damage (p value = 0.44). Field plots were treated with pesticides once every 2 weeks, which did not entirely eliminate leaf damage on the treated individuals. A more aggressive pesticide treatment regime would have been necessary to abolish leaf damage in the treated group. In addition, the levels of leaf

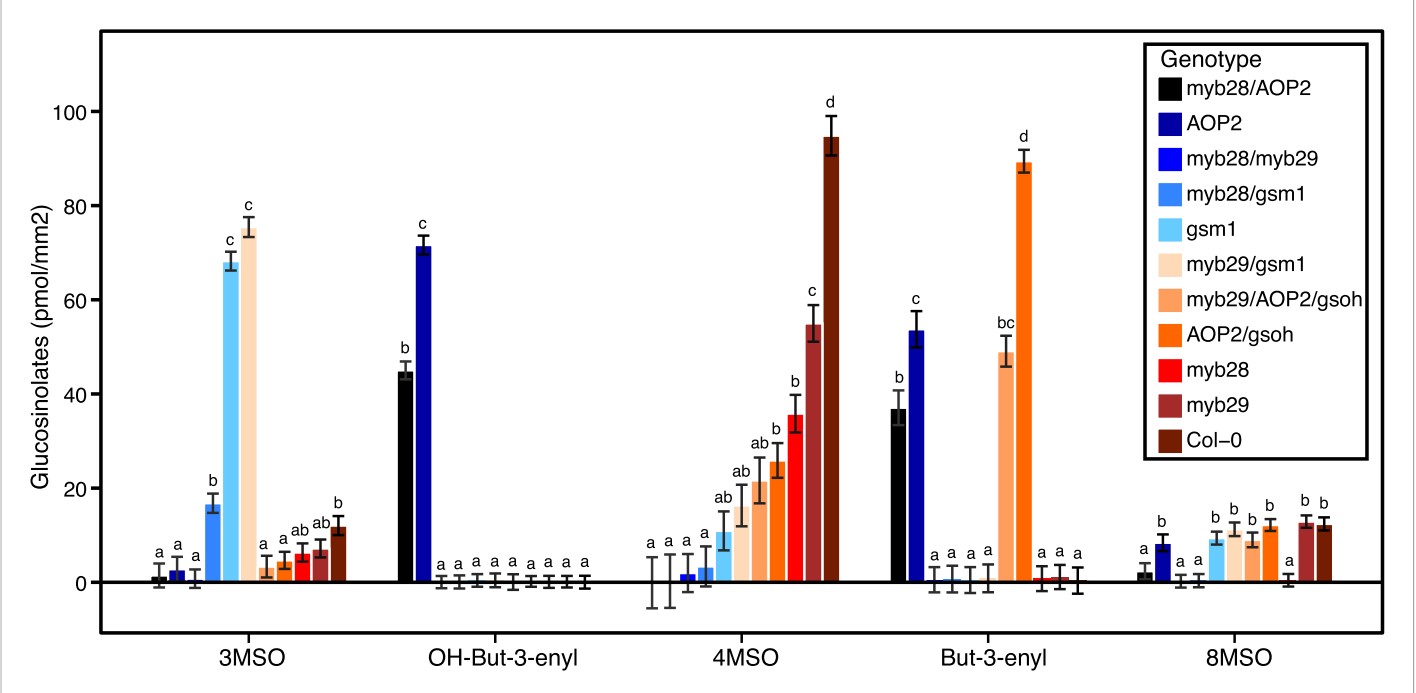

**Figure 4**. Average GSL profiles from select laboratory population genotypes grown in the field. Shown are the genotype averages of various aliphatic GSL chemical structures from GSL genotypes grown in all three environments in the field. The GSL structures present and the corresponding abundances makeup the GSL profile of an individual. Results are based on single leaf analysis of 4 week old plants (see **Table 2** for full list of GSL genotypes used in this study). Each color represents a different aliphatic GSL genotype. Error bars represent standard error of the mean. Letters represent significantly different genotypes based on Tukey's HSD. See **Figure 4—source data 1** for full list of GSL genotypes used in this study and the corresponding LSMeans and SE of GSL structures produced by all GSL genotypes used in study averaged across field trials.

The following source data is available for figure 4:

**Source data 1**. LSMeans and SE of GSL structures produced by all GSL genotypes used in study averaged across field trials.

damage measured in our study are low relative to other field studies in *Arabidopsis* (**Bidart-Bouzat and Kliebenstein, 2008**). The field site was located adjacent to other experimental field sites and greenhouses that also treated for pests, which may or may not have had an impact on the relative levels and/or pesticide resistance of herbivores in the vicinity. This combination of low overall leaf damage levels and the fact that the pesticide treatment did not eliminate leaf damage in the treated group is likely the cause for this lack of a treatment effect. However, there is a significant environment effect for leaf damage, indicating that this trait varied across the three field trials. In fact, we see no significant correlation of leaf damage across the three environments (**Table 5**). This suggests the three environments experienced differing herbivory pressures. Since we did not measure herbivore levels, we cannot determine whether the differences in leaf damage are the direct result of differences in insect populations. It is interesting to note that the UWY field site showed both the highest and lowest leaf damage levels, demonstrating that there can be potentially large temporal fluctuations in herbivory at a single location (**Table 3—source data 2**).

## Environment interacts with GSL genotype to impact leaf damage in the field

GSL variation is known to affect leaf damage incurred by insect herbivory within a controlled lab setting and we wanted to test if this could also be observed within a naturally fluctuating field setting (**Lambrix et al., 2001**; **Kliebenstein et al., 2002**; **Beekwilder et al., 2008**; **Hansen et al., 2008**). Within a field environment, levels of leaf damage significantly varied across GSL genotypes, in agreement with the role of GSL in deterring herbivory (**Table 3**). However, the extent of leaf damage incurred upon different GSL genotypes in the field fluctuated among environments, such that no

**Table 4**. PCA comparison of GSL profiles produced by GSL genotypes from synthetic laboratory population grown in the chamber and all three environments

| Environment | PCA1 = 48.5% | | PCA2 = 29% | | PCA3 = 16% | | PCA4 = 6% | |
|---|---|---|---|---|---|---|---|---|
| | R | p value | R | p value | R | p value | R | p value |
| Chamber | 1.00 | | 1.00 | | 1.00 | | 1.00 | |
| UCD2012 | 0.97 | <0.001 | 0.97 | <0.001 | 0.82 | <0.001 | 0.96 | <0.001 |
| UWY2011 | 0.91 | <0.001 | 0.95 | <0.001 | 0.74 | <0.001 | 0.97 | <0.001 |
| UWY2012 | 0.90 | <0.001 | 0.91 | <0.001 | 0.85 | <0.001 | 0.86 | <0.001 |

Glucosinolate analysis was conducted on the 17 genotypes within a Long-day growth chamber (16 hr light) set to match the median light regime for the three environmments. Principal component analysis was conducted on the mean glucosinolate accumulation for the aliphatic glucosinoles within the chamber environment. This creates a set of mathematical descriptors of the chemotype variation across the 17 genotypes. The first three eigenvectors were used to generate scores from the lsmeans of the glucosinolates across the 17 glucosinolate genotypes independently for the chamber and three different field environments values. These scores were then correlated to test if the GSL profiles were similar or not across the environments. The R of the correlation to the Chamber scores for the 17 genotypes for each of the three PCA vectors are shown in conjunction with the p value as determined by Pearson correlation. To the right of each PCA vector label is shown the fraction of total variance approximated by the given vector. In total, the four vectors describe >99% of the GSL profile variance.

particular GSL genotype showed a consistent maximal or minimal level of leaf damage across the three field trials (*Figure 6*). For example, the *myb28/AOP2* and *AOP2* genotypes had similar herbivory in UCD2012 (mean = 1.30 and 1.95, respectively) and UWY2011 (mean = 1.29 and 0.80, respectively) but strongly diverged in UWY2012 (mean = 1.45 and 5.64, respectively) (*Figure 6* and *Figure 6—source data 1*). It has been shown, in a laboratory setting that the extent to which GSL profile provides resistance varies across different herbivore species (*Kroymann et al., 2003*; *Pfalz et al., 2007*; *Hansen et al., 2008*). In addition, GSL have been shown to provide resistance to fungi, bacteria and nematodes, which may have also been present and variable between our environments (*Manici et al., 1997*; *Tierens et al., 2001*; *Aires et al., 2009*; *Witzel et al., 2013*). It is likely that the composition of the herbivore communities differed between the two field sites. Though we did not conduct a complete survey of the herbivores present at UWY and UCD, we did observe differences in leaf damage patterns between the two locations, suggesting that there would be differences in the composition of herbivores species present. Together, these results show that GSL variation controls differential leaf damage in each field trial but the specific directions of effect for individual GSL genotypes is subject to environmental conditions, such as the composition of herbivores, which can vary temporally and spatially.

## GSL variation and the environment impact fitness in the field

Since our laboratory population contains single gene variants, we have the ability to test the fitness consequences of individual genotypes in a field setting, an important step in connecting the GSL variation observed among *Arabidopsis* accessions with potential selective and non-selective evolutionary processes. To test if the GSL genotypes alter plant fitness in the three environments, we measured fecundity of each individual grown in the field. Plants were harvested from the field at maturity and the numbers of fruits, flowers and buds per plant were counted in the laboratory to yield total fruit count (TFC). TFC has previously been shown to be a good proxy for fecundity in *Arabidopsis* where total number of seeds per plant is highly correlated with total number of siliques (i.e., fruits) (*Wolf et al., 2000*; *Kliebenstein et al., 2001c*). Among the GSL genotypes we observed variation in silique length. *Arabidopsis* siliques contain two rows of seeds in a linear conformation, so that silique length strongly correlates with seed number at maturity, assuming uniform seed size. Therefore variation in silique length or seed size could affect our fecundity estimates. Silique length and seed size were measured from digital images of GSL genotypes harvested from the field and seed size showed no significant variation (data not shown). However, there was significant variation in silique length across GSL genotypes as well as a significant genotype by environment interaction (*Table 3—source data 1*). We concluded that the significant differences in silique lengths are likely

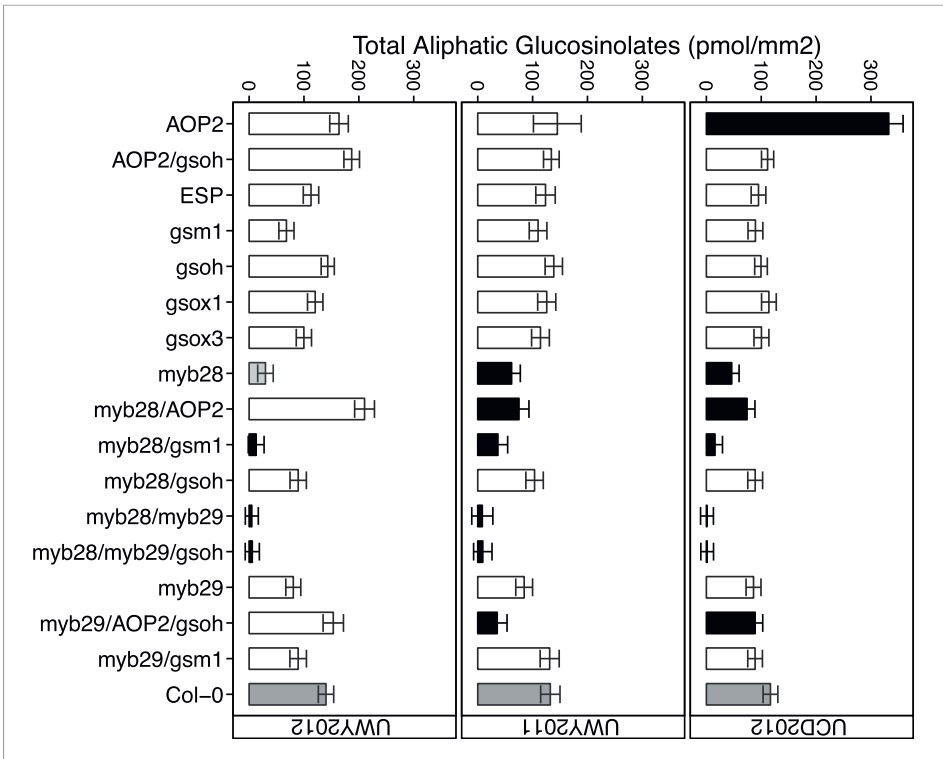

**Figure 5**. Total aliphatic GSL accumulation of GSL genotypes from the laboratory population grown in the field. Shown are the genotype averages in all three environments of total aliphatic GSL from individuals grown in the field. Results are based on single leaf analysis of 4 week old plants. Bar color based on Dunnett's multiple comparison procedure. Within each environment, dark grey bars = Col-0 genotype, black bars = genotypes that accumulate significantly more or less total aliphatic GSL than Col-0 (p value ≤ 0.05), light grey bars = genotypes that accumulate suggestively more or less total aliphatic GSL than Col-0 (p value = 0.05–0.1) and white bars = genotypes that are not significantly different than Col-0 (p value >0.1). Error bars represent standard error of the mean.

reflective of fecundity and adjusted our fitness measurements using this information. Estimates of absolute fitness were therefore obtained for each individual as TFC multiplied by silique length both including and excluding individuals that did not survive to harvest. Survivorship was included in fitness estimates to avoid obtaining artificially inflated fitness estimates from GSL genotypes with higher death rates that would result from removing individuals that do not survive to fruiting and have a fitness of zero.

In this study, GSL genotype had a significant impact on absolute field fitness (*Table 6*). There was also a significant interaction effect between GSL genotype and environment for absolute fitness both including and excluding survivorship, suggesting that the impact that GSL genotype has on fitness is conditioned upon the environment (*Table 6*). Environment did not show a significant main effect on either measure of absolute fitness, suggesting that the population mean for absolute fitness may have been comparable across the environments and instead it is the fitness of GSL genotypes relative to each other within an environment that varies. Thus, these GSL genotypes that recreate natural variation within a single common genetic background influence field fitness of *A. thaliana* in an environmentally dependent manner.

To visualize if the rank in absolute fitness of GSL genotypes fluctuates among the three environments and to compare the patterns of fluctuation of GSL genotypes across environments, we plotted the mean normalized fitness of all GSL genotypes in all environments for both absolute fitness measures, including and excluding survivorship (*Figure 7* and *Figure 7—figure supplement 1*). Absolute fitness varied greatly between the highest and lowest ranked GSL genotypes within each of the environments (*Figure 7* and *Figure 7—source data 1*). In addition, the performance of different GSL genotypes relative to each other varied across environments, so that no GSL genotype

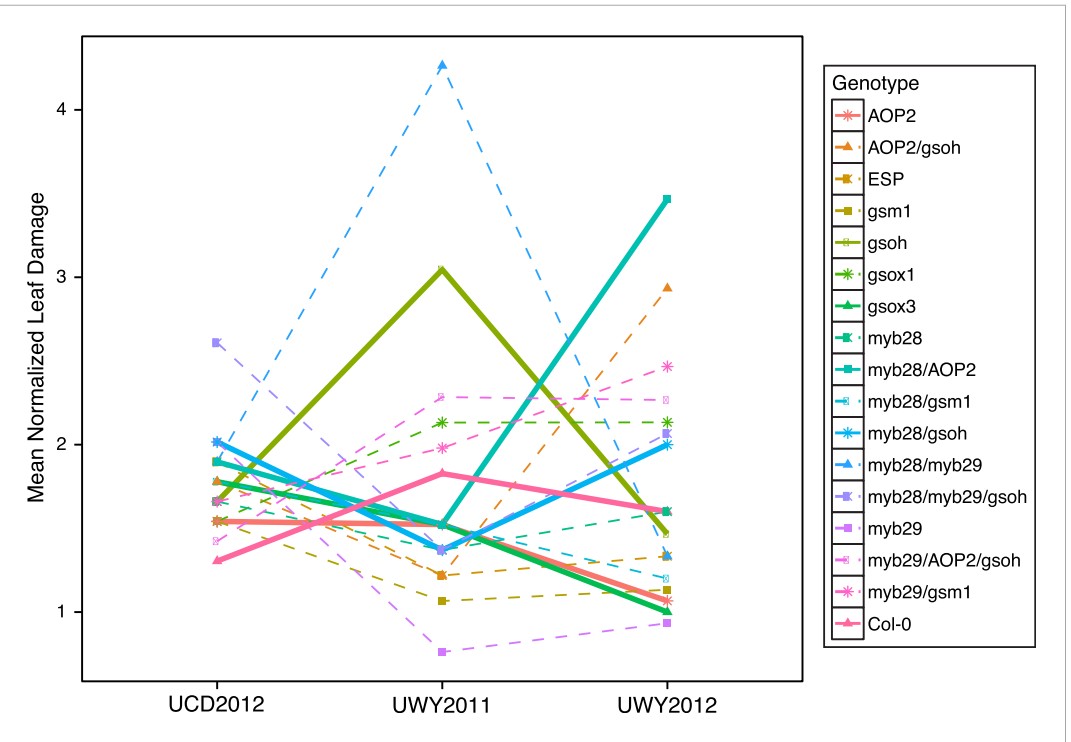

**Figure 6**. Mean normalized leaf damage of GSL genotypes from the laboratory population grown in the field. Shown are the mean normalized genotype averages in all three environments of leaf damage from GSL genotypes grown in the field. Mean normalization was conducted by first dividing the genotype average of each GSL genotype within an environment to the corresponding environment average. Then, each normalized value was multiplied by the grand mean across all three environments. This was done in order put the leaf damage estimates in each environment on the same order of magnitude to ease visual comparisons of genotypes across environments and to highlight the fact that relative leaf damage of a given GSL genotype varies across environments.

The following source data is available for figure 6:

**Source data 1**. Mean normalized values for leaf damage in the field.

outperformed all the others in all three environments. For example, *myb28/AOP2* shows the greatest fitness in the UCD2012 environment and the lowest fitness in UWY2012. In contrast, *myb28/gsoh* shows an opposite pattern while other genotypes showing a diversity of other patterns (*Figure 7*). This fluctuation in rank of GSL genotypes across environments can also be observed if we look at fluctuations of TFC with and without survivorship across the three environments, though the patterns for specific GSL genotypes vary across the different fitness measures (*Figure 7—figure supplement 1*). Thus, it appears that the significant interaction of GSL genotype by environment controlling fitness is caused by fluctuations in the fitness rank of different genotypes across environments (*Figure 7* and *Figure 7—source data 1*).

**Table 5**. Environmental correlations for leaf damage in the field

| UCD2012-UWY2012 | | UWY2012-UWY2011 | | UCD2012-UWY2011 | |
|---|---|---|---|---|---|
| R | p value | R | p value | R | p value |
| −0.25 | – | 0.01 | – | 0.22 | – |

Shown are Pearson's correlations for leaf damage between the different environments. R = correlation coefficients; p value = probability statistic. Non-significant p values are represented by a dash.

**Table 6**. Mixed model of fitness phenotypes in the field

| Fixed effects | Absolute fitness (w/survivorship) | | | | | Absolute fitness (w/out survivorship) | | | | |
|---|---|---|---|---|---|---|---|---|---|---|
| Source of variation | df | SS | MS | F | p value | df | SS | MS | F | p value |
| Genotype | 16 | 455453 | 28466 | 2.2 | 4.9E-03 | 16 | 397186 | 24824 | 2.3 | 2.0E-03 |
| Environment | 2 | 88326 | 44163 | 2.2 | – | 2 | 127177 | 63588 | 4.6 | – |
| Treatment | 1 | 1948 | 1948 | 0.2 | – | 1 | 2042 | 2042 | 0.3 | – |
| Geno:Env | 32 | 706781 | 22087 | 1.7 | 0.01 | 32 | 508962 | 15905 | 1.5 | 0.04 |
| Geno:Trt | 16 | 137795 | 8612 | 0.6 | – | 16 | 169036 | 10565 | 1.0 | – |
| Env:Trt | 2 | 2918 | 1459 | 0.1 | – | 2 | 4883 | 2442 | 0.2 | – |
| Geno:Env:Trt | 32 | 348291 | 10884 | 0.8 | – | 32 | 235224 | 7351 | 0.7 | – |
| **Random Effects** | **Absolute fitness (w/survivorship)** | | | | | **Absolute fitness (w/out survivorship)** | | | | |
| Source of variation | df | SS | MS | Chi.sq | p value | df | SS | MS | Chi.sq | p value |
| Plot(Trt:Env) | 1 | 2640 | 2640 | 216.5 | 0 | 1 | 3311 | 3311 | 279.3 | 0 |
| Residual | 1692 | 12581 | 7 | NA | – | 1451 | 10061 | 7 | NA | – |
| **Fixed effects** | **Relative fitness** | | | | | **Survivorship** | | | | |
| Source of variation | df | SS | MS | F | p value | df | SS | MS | F | p value |
| Genotype | 16 | 28 | 2 | 2.7 | 3.9E-04 | 16 | 5 | 0 | 4.9 | 6.9E-10 |
| Environment | 2 | 2 | 1 | 1.0 | – | 2 | 2 | 1 | 9.3 | 0.01 |
| Treatment | 1 | 0 | 0 | 0.2 | – | 1 | 0 | 0 | 1.2 | – |
| Geno:Env | 32 | 39 | 1 | 1.8 | 4.2E-03 | 32 | 9 | 0 | 3.8 | 3.8E-12 |
| Geno:Trt | 16 | 8 | 1 | 0.8 | – | 16 | 1 | 0 | 0.8 | – |
| Env:Trt | 2 | 0 | 0 | 0.1 | – | 2 | 0 | 0 | 0.1 | – |
| Geno:Env:Trt | 32 | 17 | 1 | 0.8 | – | 32 | 3 | 0 | 1.1 | – |
| **Random effects** | **Relative fitness** | | | | | **Survivorship** | | | | |
| Source of variation | df | SS | MS | Chi.sq | p value | df | SS | MS | Chi.sq | p value |
| Plot(Trt:Env) | 1 | 0.1 | 0.1 | 184.5 | 0 | 1 | 0 | 0 | 0 | – |
| Residual | 1692 | 1 | 3.8E-04 | NA | – | 1900 | 0.1 | 3.5E-05 | NA | – |

A linear mixed model was fitted to phenotypes measured on plants grown in the field. Variation was partitioned among the fixed effects, Genotype, Environment, and Treatment as well as a random factor, Plot, inside which Treatment and Environment were nested. Phenotypic data used in the model was collected on 17 genotypes from three environments and two treatment groups. df = degrees of freedom, SS = Type II Sums of Squares variation, MS = Mean Squared variation, F = F statistic. Non-significant p values (>0.05) are represented by a dash.

Within an environment, individuals compete against their neighbors for resources during their lifetime and natural selection favors those with higher performance relative to others. Therefore, in addition to absolute fitness, we also analyzed the effect of the GSL genotype on relative fitness in the field, both with and without survivorship. We calculated relative fitness of each GSL genotype within each environment as absolute fitness divided the population mean within that environment. Even more strongly than with our absolute fitness measurements, we found that GSL genotype and the interaction between GSL genotype and environment both had a significant impact on relative fitness in the field both including and excluding survivorship (*Table 6*). For example, myb28 has a higher than average relative fitness in UWY2011 but shows an average and slightly lower than average relative fitness in UWY2012 and UCD2012, respectively (*Figure 8*). In other cases, relative fitness of a GSL genotype is similar among the UWY field trials but differs in the UCD field trial. Two examples, with opposite patterns are *myb28/AOP2*, that has low relative fitness in both UWY field trials but higher relative fitness in UCD and gsm1, that has high relative fitness in both UWY field trials but lower relative fitness in UCD. This indicates that temporal and spatial fluctuations in fitness can both occur and are dependent on genotypic differences.

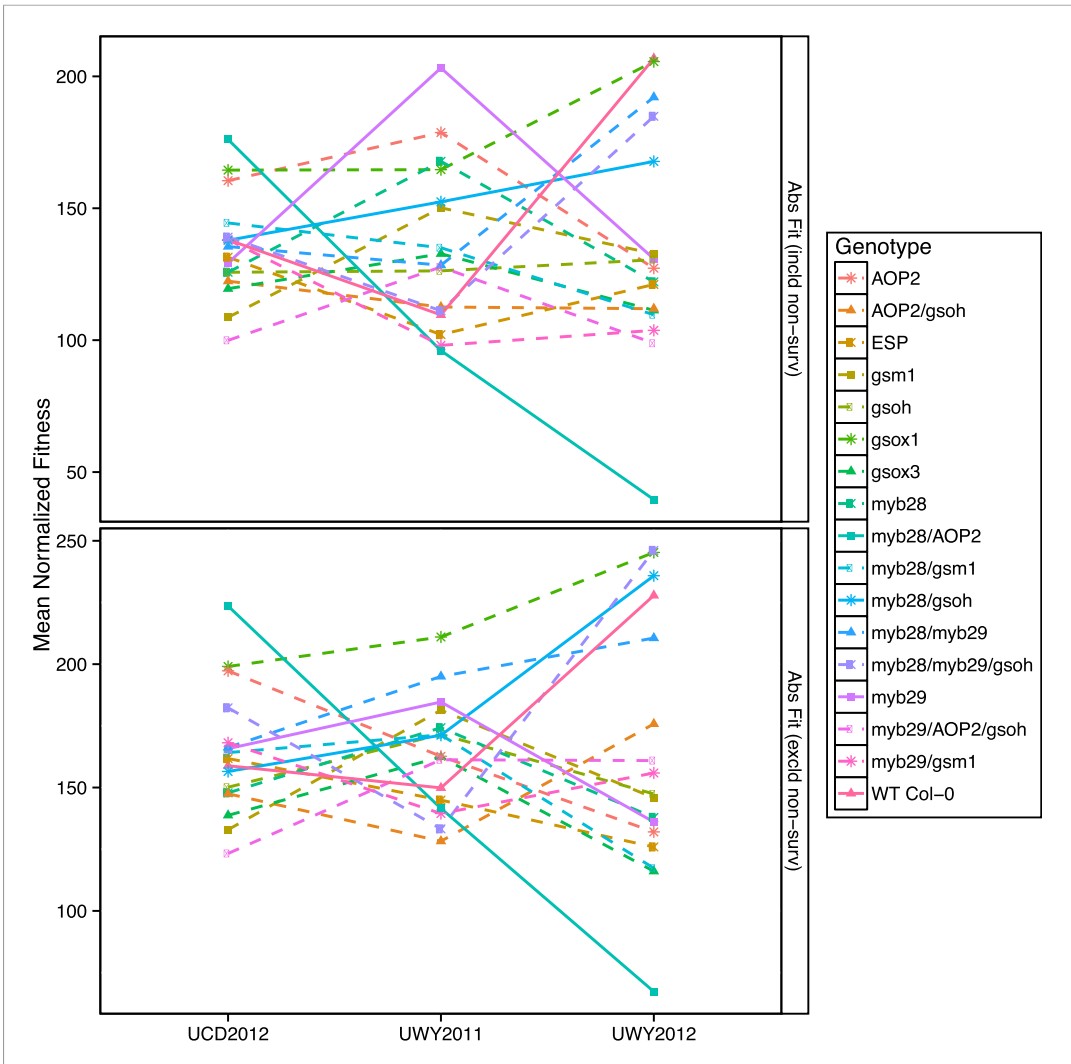

**Figure 7**. Mean normalized absolute fitness of GSL genotypes from the laboratory population grown in the field. Shown are the mean normalized genotype averages of absolute fitness from GSL genotypes grown in all three environments calculated either including or excluding survivorship, as indicated. Absolute fitness including survivorship was calculated as total fruit count (TFC) × silique length × survivorship, whereas absolute fitness excluding survivorship was calculated as TFC × silique length for individuals that survived to harvest. Mean normalization was conducted for each phenotype by first dividing the average of each GSL genotype within an environment to the corresponding population mean for each environment. Then, each normalized value was multiplied by the grand mean across all three environments. This was done in order put the phenotype estimates in each environment on the same order of magnitude to ease visual comparisons. Solid lines represent distinct patterns that GSL genotypes display across the environments and are meant as a visual aid.

The following source data and figure supplements are available for figure 7:

**Source data 1**. Mean normalized values for phenotypic traits in the field.

**Figure supplement 1**. Mean normalized total fruit count of GSL genotypes from the laboratory population grown in the field.

Interestingly, heatmaps of absolute fitness and relative fitness reveal unexpected hierarchical clustering of the environments between the two traits (*Figure 8*). In both cases, UCD2012 clusters with UWY2011 and the two UWY field trials do not cluster together, showing that within an environment across years there is the potential for greater variability than across environments.

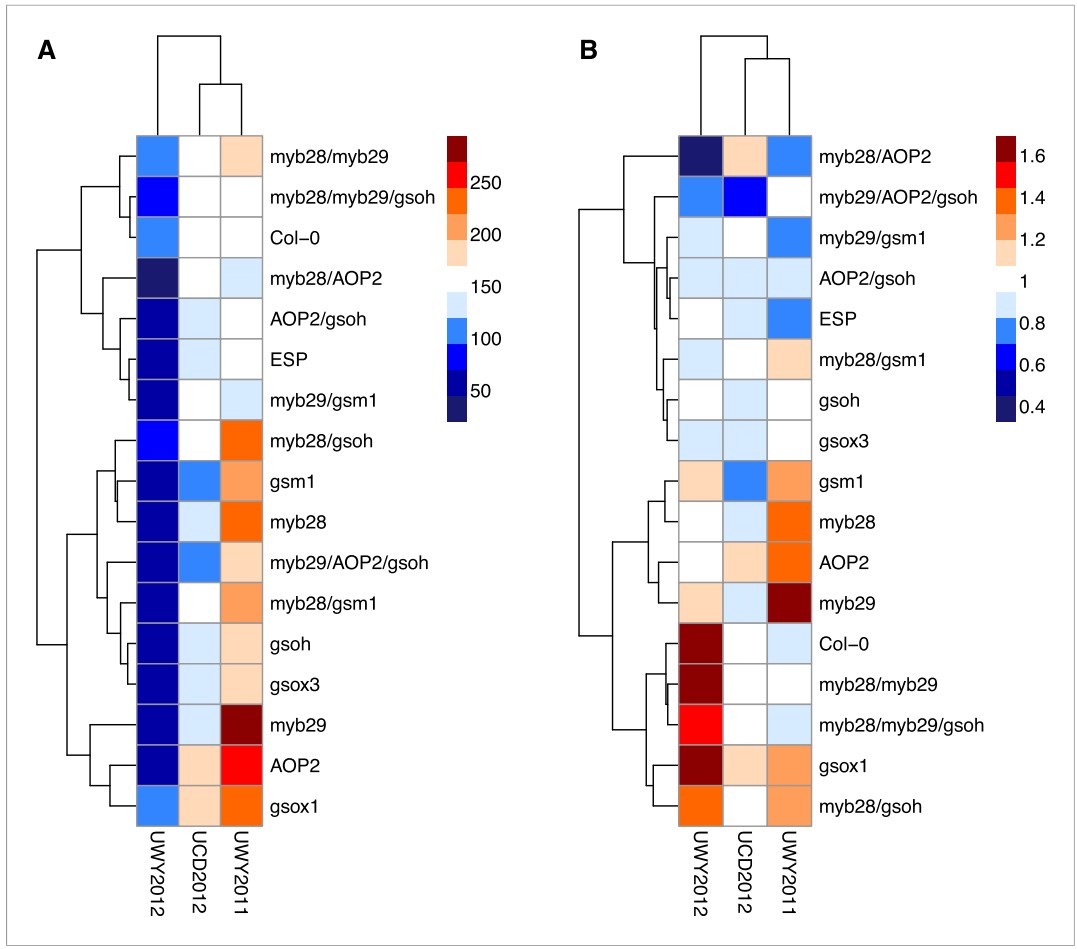

**Figure 8**. Relative and absolute of GSL genotypes from the laboratory population grown in the field. Heatmaps with hierarchical clustering of GSL genotypes representing the model corrected means of (**A**) absolute fitness including survivorship and (**B**) relative fitness of each genotype in each environment. Absolute fitness was calculated for each individual as the total fruit count × silique length × survivorship. Relative fitness was calculated by normalizing absolute fitness for each genotype against the population mean within an environment.

## Pleiotropic links to GSL genes

In our analysis, we measured flowering time and total indole GSL in the field. In a laboratory setting, GSL genes have been observed to pleiotropically alter these traits (*Kerwin et al., 2011*). In the field, both of these phenotypes were significantly affected by the GSL genetic variation in our synthetic population, indicating that aliphatic GSL genes can have pleiotropic impacts beyond the aliphatic GSL pathway that can be observed in natural settings (*Table 3* and *Table 3—source data 1*). Therefore, there is the possibility that either of these phenotypes might be driving the observed variation in fitness of GSL genotypes across these environments. To test this, we conducted genetic correlations using the genotypic means for absolute fitness, flowering time and total indole GSL within each environment (*Table 7*). We did not observe a significant correlation between absolute fitness and our pleiotriopic traits, using either parametric or non-parametric approaches, in any of our three environments (*Table 7*). This indicates that while the GSL genes are causing pleiotropic effects, these pleiotropic effects are probably not driving the observed fitness consequences of the GSL genotypes in our field trials.

## Non-random variation of GSL loci among field collected accessions

To test if natural *Arabidopsis* accessions show a pattern of variation consistent with fluctuating selection, we determined the GSL haplotype for a global collection of accessions using their GSL

Table 7. Genetic correlations between fitness and Pleitropic traits in the field

| | Absolute fitness | Flowering time | Total indole GSL |
|---|---|---|---|
| Trait (UCD2012) | | | |
| Absolute fitness | – | −0.40 | −0.21 |
| Flowering time | −0.22 | – | 0.06 |
| Total indole GSL | −0.05 | −0.23 | – |
| Trait (UWY2011) | | | |
| Absolute fitness | – | −0.24 | −0.27 |
| Flowering time | −0.19 | – | −0.05 |
| Total indole GSL | −0.43 | 0.12 | – |
| Trait (UWY2012) | | | |
| Absolute fitness | – | 0.22 | −0.25 |
| Flowering time | 0.29 | – | 0.59* |
| Total indole GSL | −0.21 | 0.38 | – |

Shown are genetic correlations between absolute fitness and traits pleiotropically controlled by GSL genes. Pearson's correlation coefficients are on the top half of the tables and Spearman rank correlations are on the bottom. *p value < 0.05, **p value < 0.01, ***p value < 0.001.

profile (Figure 2). Using the validated GSL phenotype caused by genetic variation at the eight causal genes for the aliphatic GSL pathway, we assigned a GSL haplotype to each *Arabidopsis* accession, given its GSL profile (Table 1 and Figure 1). Using the available GSL profile information, the underlying allelic state at each of the eight genes assigned functional or non-functional, based on presence or absence of different GSL structures as well as the relative abundances of different structures, that is, based on the GSL profile of the accession. This identified 18 distinct aliphatic GSL haplotypes among the set of 144 natural *Arabidopsis* accessions, observed at different frequencies (Figure 9 and Figure 9—source data 1). Using the observed single locus allelic frequencies, we calculated the expected GSL haplotype frequencies for each of the 18 multi-locus genotypes (Figure 9—source data 1). These expected frequencies for the GSL genotypes represent theoretical frequencies that would be expected if no selection gradient acted upon GSL variation and no genetic drift, migration or other non-selective effect upon population structure biased the allele distribution. Comparing the population of observed vs expected frequencies was highly non-random (p < 0.001) (Figure 9 and Figure 9—source data 1). Further, specific multi-locus GSL genotypes occurred significantly more or less often than expected (Figure 9 and Figure 9—source data 1). Thus, the non-random variation of GSL haplotypes within the *Arabidopsis* accessions supports the observations from the empirical field trials. It is similarly possible that this observed non-random variation is caused by non-selective processes such as migration, population structure and/or local bottleneck. Significant future efforts will be required to test the extent to which this non-random variation is caused by neutral demographic processes vs potential fluctuating selection.

## Discussion

Ecologically and evolutionarily important traits often show considerable phenotypic variation in nature that is quantitative, polygenic and interacts with the environment. A clear example of this is aliphatic GSL accumulation in *Arabidopsis*, which is highly polygenic and environmentally dependent (Figures 1, 4, 5). However, it has been complicated to validate that specific polymorphic loci within a pathway are the actual causative basis of any changes in fitness due to the use of polygenic populations (Lande and Arnold, 1983). In this study, using a single gene manipulation approach that has allowed us, over the past decade, to recreate natural allelic diversity in the aliphatic GSL pathway, we have shown that GSL genetic variation at numerous loci directly impacts *Arabidopsis* fitness in the field (Table 1, Figure 7, Figure 7—figure supplement 1, Figure 8). Because we have only manipulated the GSL genes within an otherwise isogenic background, we can directly conclude that it is these specific genes and their GSL phenotypes that are determining the differences in fitness in the field. Further experiments will optimally generate the full 256-line matrix containing all combinations of alleles

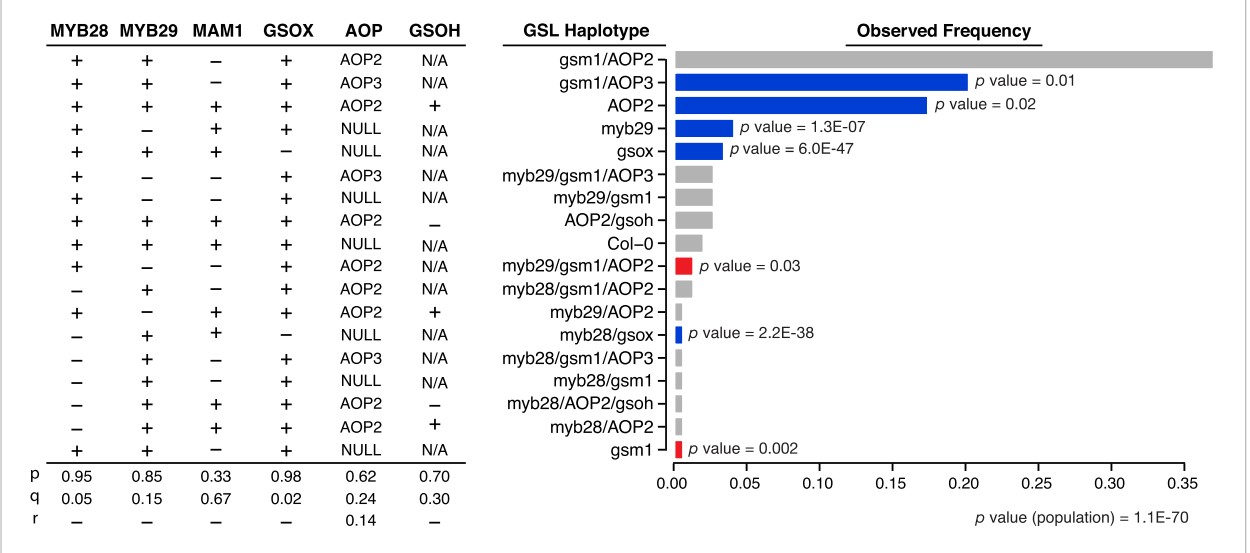

**Figure 9**. GSL haplotype frequencies of *Arabidopsis thaliana* accessions based on GSL profile data from chamber-grown individuals. Shown are the GSL haplotypes observed among a population of 144 *Arabidopsis* accessions for which our lab had existing GSL data. Seven loci important in the aliphatic GSL pathway were called based on GSL profile data from the lab as '+' = functional, '−' = non-functional or 'NA' = unobservable due to epistasis (see *Figure 1* for an explanation of epistasis in the GSL biosynthetic pathway). Bar length represents the observed GSL genotype frequencies among 144 *Arabidopsis* accessions. Bar color represents the difference, for a given GSL haplotype, between expected and observed genotype frequencies, based on Chi Squared distribution (significant p values shown). Blue = GSL genotypes found more frequently than expected (p value ≤ 0.05), red = GSL genotypes found less frequently than expected (p value ≤ 0.05) and grey = GSL genotypes found as frequently as expected (p value >0.05).

The following source data is available for figure 9:

**Source data 1**. χ2 analysis comparing observed and expected GSL haplotype frequencies observed among a globally distributed population of 144 *Arabidopsis thaliana* accessions.

between all loci to fully interrogate the effects of all loci in all possible backgrounds. We should also note that even with all of our efforts to clean up the respective backgrounds and validate that the mutant phenotypes are similar to the segregating natural genotypes, it remains possible that some of the observed effects are caused by unexpected changes in the lines.

More difficult however is to ascribe the specific selective forces acting on this GSL variation to produce a fitness effect. GSL are known plant defensive compounds and variation in GSL genotype was shown to significantly impact GSL profiles, leaf damage and fitness in the field (*Table 3*). While GSL variation did alter measured leaf damage in the field, the patterns did not fully reflect the relative fitness spectrum of these same genotypes (*Figures 6–8*). One possibility is that our experiment, even with 20 blocks (10 control/10 pesticide treated) per field trial, was still insufficient to identify the underlying link, suggesting the need for larger experiments. Another possibility is that there were different herbivore populations between these environments, which agrees with the observation that there was no genetic correlation of herbivore resistance across the three field trials (*Table 5*). The fact that different GSLs defend against different herbivores would complicate finding the specific link between GSL loci and a population of herbivores (*Kroymann et al., 2003*; *Falk and Gershenzon, 2007*; *Pfalz et al., 2007*; *Hansen et al., 2008*; *Falk et al., 2014*). Additionally, our herbivory measures are limited to foliar damage, which obfuscates any potential interactions between GSL genotype and root pathogens. Supporting this idea, previous studies have found that GSL can influence a number of root pathogens and commensal microbes (*Bending and Lincoln, 2000*; *van Dam et al., 2008*; *Bressan et al., 2009*; *Millet et al., 2010*; *Witzel et al., 2013*). While these organisms could directly impact plant fitness, this interaction is highly difficult to detect or control in field trials.

In addition to unmeasured biotic stresses, there is the potential for causal links between GSL genes and abiotic pressures. We showed that the GSL genes have pleiotropic effects on development such as flowering time that while having no link to fitness in our experiments could impact fitness in other

environments. Similarly, previous work has shown that individual GSL structures directly modulate stomatal closure in response to wounding (*Zhao et al., 2008*). Furthermore, analysis of natural variation and validation lines showed that GSL structure and amount can influence the circadian clock and flowering time (*Kerwin et al., 2011*). Other experiments have also identified a potential for regulatory roles with indole GLS (*Clay et al., 2009*). Thus, these are not indirect pleiotropies but direct regulatory links whereby GSLs may influence the plants abiotic responses potentially to alter the biotic interactions. Thus, it is possible that the observed GSL to fitness links are resulting from a complex web of biotic and abiotic effects. Identifying the specific selective agents affected by GSL variation will require the development of techniques for rapid and systematic identification of all foliar and root herbivores and microbes from field samples as well as a complete physiological and developmental analysis of the plant within the field. This is especially critical as the specific agents of selection may be highly variable across environments.

Within our multiple field trials, we found that effects of GSL genes on fitness are highly dependent upon the environment in which the experiment is conducted (*Table 6*). The fitness effects of the naturally polymorphic GSL genes were such that each environment had a different optimal set of GSL genotypes (*Figure 7*, *Figure 7—figure supplement 1*, Figure 7—figure supplement 8). Similarly, no particular GSL genotype had the maximal fitness in all environments (*Figure 7*, *Figure 7—figure supplement 1*, *Figure 8*). This suggests that the GSL defense pathway might be a system in which genetic variation could be stabilized by fluctuating selection across the environments. Fully exploring this hypothesis will require extensive assessment of genetic variation at the polymorphic GSL loci within natural populations and more extensive field trials of this synthetic population that recreates natural diversity at these same loci.

Within species that are highly but not exclusively selfing, such as *A. thaliana*, temporal variation in selection is not solely sufficient to maintain genetic diversity (*Dempster, 1955*; *Bomblies et al., 2010*). This would require either spatial variation in fitness and/or variation within a seed bank to provide extra drive for the system (*Dempster, 1955*; *Turelli et al., 2001*; *Turelli and Barton, 2004*). Recent work has begun to show that *Arabidopsis* has a robust multi-generational seed bank in natural populations (*Lundemo et al., 2009*; *Bomblies et al., 2010*). Further, there is extensive allelic variation within small local regions that contain different habitats, that would likely experience different insect pressures, providing the potential for spatial variation in fitness (*Bomblies et al., 2010*). Thus, both conditions necessary for fluctuating selection to maintain diversity in *Arabidopsis* exist, but we do not yet know enough about the extent of the seed bank or spatial variation in selection within *Arabidopsis* to fully model the system. This shows that a greater understanding of life history traits, seed bank history and migration rates in natural populations of *Arabidopsis* is necessary to determine if fluctuating selection is contributing to the maintenance of variation in this species.

## Conclusions

Based on our measures of fitness in the field, we showed that GSL variation can control fitness within the field. These fitness effects were not driven by pleiotropic phenotypes like flowering, but the specific selective pressures driving these fitness differences remain to be identified. Identifying these pressures will require vastly larger surveys of natural populations and long-term field trials. Using the empirical values for fitness, we could show that the GSL system within these environments fits models where fluctuating selection can maintain standing polygenic variation. Further trials are required to test if this is more broadly applicable across a broader range of environments. This would require more field trials using our synthetic population to provide the capacity to empirically evaluate models of maintenance of standing variation and its influence on adaptation (*Gillespie and Turelli, 1989*; *Orr, 1998*; *Agrawal, 2001*). It remains to be directly tested if similar evolutionary processes drive evolution of other ecologically important traits that must respond to fluctuating environmental conditions such as pathogen populations and water availability.

## Materials and methods

### Synthetic laboratory population generation

The following eight loci in the aliphatic GSL pathway were modified in the synthetic laboratory population of *A. thaliana* genotypes: *AOP2* (At4g03060), *ESP* (At1g54040), *MYB28* (At5g61420), *MYB29* (At5g07690), *GSOH* (At2g25450), *MAM1* (At5g23010), *GSOX1* (At1g65860), *GSOX3* (At1g62560).

The following knockout or complementation lines for the following loci in *A. thaliana* Col-0 were used to generate the lab population: *AOP2* = 35S:AOP2 (*Li and Quiros, 2003*), *ESP* = 35S:ESP (*Burow et al., 2006*), *MYB28* = SALK_136312, (*Sønderby et al., 2007*), *MYB29* = SM.34316 (*Hirai et al., 2007*), *GS-OH* = SALK_09807 (*Hansen et al., 2008*), *MAM1* = EMS mutant line gsm1 (*Haughn et al., 1991*), *GSOX1* = SALK_079493 (*Li et al., 2008*), *GSOX3* = CSHL_GT13906 (*Li et al., 2008*). Mutant lines were manually crossed to each other to generate a population of plants containing homozogyous combinations of mutations in the different genes mentioned above, representing a subset of the potential variation in the aliphatic GSL pathway observed among *Arabidopsis* accessions (*Table 2*). Individuals were genotyped via PCR using the primers and reaction conditions listed below.

## Experimental settings

Field trials were conducted in two locations, the latter over 2 years, giving three separate environments total. The first field trial was performed at the University of Wyoming (UWY) in Laramie, WY during Summer 2011, the second at UC Davis in Davis, CA Spring 2012, and the third at UWY during Summer 2012. Seeds were sown into flats with 2 inch 50-celled inserts using Sunshine #5 (Sungro, Agawam, MA) potting soil containing slow release fertilizer and stratified at 4°C for 4 days before being transferred into the greenhouse at either the University of Wyoming in Laramie (UWY) or the University of California at Davis (UCD) to facilitate germination synchrony. In the UWY greenhouse, plants received 15 hr light/9 hr dark natural phototoperiod with temperatures fluctuating diurnally from 10°C to 30°C. In the UCD greenhouse, plants received 14 hr light/10 hr dark natural photoperiod with temperatures fluctuating from 15°C to 35°C. Further, starting all the plants in the greenhouse minimizes variation in the initial seedling conditions. After germination, seedlings were thinned to one per pot and GSL genotypes were randomly organized into 40 blocks per field trial, for a total of 120 blocks total and also 120 GSL genotype replicates total. Individuals were transplanted

PCR primer sets and reaction conditions for genotyping

| Locus | Primers sequence | Group |
|---|---|---|
| MYB29 gene | myb29-1 RP 5'-TATGTTTGCATCATCTCGTCTTC-3' | 1 |
| | myb29-1 LP 5'-TTGTAGATTGCGATGGGCTA-3' | |
| MYB29 T-DNA | myb29-1 RP 5'-TATGTTTGCATCATCTCGTCTTC-3' | 1 |
| | myb29-1 LB 5'-ATATTGACCATCATACTCATTGC-3' | |
| AOP2 gene | AOP2 FOR ODD13 5'-AACAGCGAAACGATCCAGAAGA-3' | 1 |
| | AOP2 REV ODD24 5'-GTGCTTCTCGTCCACAA-3' | |
| MAM1 gene | gsm1-2 FOR 5'-TCATCGCTTCTGACATCTTCC-3' | 1 |
| | gsm1-2 REV 5'-GTCTTGGCGATGGTCTTAATG-3' | |
| GSOX3 gene | gsox3 RP (P3P) 5'-TCGTCCTGACAAGACTGCTG-3' | 2 |
| | gsox3 LP (P3P) 5'-GAGGGTCCAGTCGAAAAACTC-3' | |
| GSOX3 T-DNA | gsox3 RP (P3P) 5'-TCGTCCTGACAAGACTGCTG-3' | 2 |
| | LB2 5'-GCTTCCTATTATATCTTCCCAAATTACCAATACA-3' | |
| GSOH gene | GSOH RP1 5'-GCTTCGGGATTAGGAGGAAC-3' | 2 |
| | GSOH LP 5'-ATGAAGATTGGCGTGAAAGG-3' | |
| GSOH T-DNA | GSOH RP1 5'-GCTTCGGGATTAGGAGGAAC-3' | 2 |
| | LBb1.3 5'-ATTTTGCCGATTTCGGAAC-3' | |
| GSOX1 gene | gsox1 RP (P3P) 5'-CTAGCGCGGGTAGAAAGACAT-3' | 3 |
| | gsox1 LP (P3P) 5'-GCATTCCAAAAATACCATAACG-3' | |
| GSOX1 T-DNA | gsox1 RP (P3P) 5'-CTAGCGCGGGTAGAAAGACAT-3' | 3 |
| | LB2 5'GCTTCCTATTATATCTTCCCAAATTACCAATACA-3' | |
| MYB28 gene | myb28-1 RP 5'-TGTATAAACCAGCTTTTTGGGG-3' | 3 |
| | myb28-1 LP 5'-TTTTTCATTATGCGTTTGCAG-3' | |
| MYB28 T-DNA | myb28-1 RP 5'-TGTATAAACCAGCTTTTTGGGG-3' | 3 |
| | LBa1 5'-TGGTTCACGTAGTGGGCCATCG-3' | |

**Reaction conditions for group 1**

| Initial melting | 32 cycles | | | Final extension |
|---|---|---|---|---|
| 94℃ | 94℃ | 60℃ | 72℃ | 72℃ |
| 30 s | 30 s | 45 s | 90 s | 10 min |

**Reaction conditions for group 2**

| Initial melting | 30 cycles | | | Final extension |
|---|---|---|---|---|
| 94℃ | 94℃ | 61℃ | 72℃ | 72℃ |
| 30 s | 30 s | 45 s | 90 s | 10 min |

**Reaction conditions for group 3**

| Initial melting | 30 cycles | | | Final extension |
|---|---|---|---|---|
| 94℃ | 94℃ | 65℃ | 72℃ | 72℃ |
| 45 s | 45 s | 45 s | 90 s | 10 min |

from the greenhouse into the field 2 weeks post germination. A single plant of each genotype was present in each block in all three environments and blocks were arranged into four rows of ten blocks each (*Figure 3*). Each row of 10 blocks is referred to as a plot, so that there were four plots per field trial and 12 plots total. Within each plot is nested a treatment by environment combination. Every 14 days, two plots (20 blocks total) per environment were treated with pesticides to decrease leaf damage due to herbivory. At UWY, plants were sprayed with the insecticide Sevin (GardenTech, Palatine, IL) to repel flea beetles. At UCD, plants were treated with Marathon 1% granular (OHP, Mainland, PA) and Lily Miller Slug, Snail & Insect Killer Bait (Lily Miller Brands, Walnut Creek, CA). The plants were allowed to grow in the field for 4 weeks before being harvested. At harvest, the aerial portion of each plant was collected from the field, placed into a quart sized freezer bag and transferred into 4℃ for temporary storage. After the harvest completion, the UCD field plants were immediately placed into −80℃ for storage. The UWY field plants were shipped to UC Davis overnight on dry ice and then placed in −80℃ for storage.

## GSL extraction, HPLC separation and GSL structure identification

GSL were measured on all field trial plants to assess field effects of the genotypes on GSL accumulation. At approximately 4 weeks of age, a single, fully expanded, green leaf was collected from each plant. In order to measure leaf area of each sample, leaves from twelve plants at a time were placed on a white sheet of paper with a grid overlay. A ruler was placed on the sheet of paper below the leaves and digitally imaged using a Nikon D3100 (Tokyo, Japan). The photographed leaves were then placed directly into 96 deep well plates containing 400 µl 90% methanol and stored in the freezer until extraction. For the UWY field trial, the leaves were stored at −20℃ for 3–4 weeks and shipped overnight to Davis, CA on dry ice. For the Davis field trial, all plates were stored at −20℃ until extraction. After harvest, desulfoglucosinolates were extracted from all samples using a high-throughput protocol briefly described below (*Kliebenstein et al., 2001a*). One gram of Sephadex DEAE A-25 (Sigma–Aldrich, St. Louis, MO) was added to each well of a 96 well filter plate using a column loader. To hydrate the Sephadex, 300 µl of $H_2O$ was transferred to each well using a multichannel pipet and allowed to incubate at room temp 1 hr. Excess $H_2O$ was removed from the Sephadex by placing filter plate on top of a 96 deep well discard plate (used to catch the flow through) and centrifuged at 1000 rpm for 2 min. To homogenize the samples, 96 deep well plates containing a single *A. thaliana* leaf, two 2.3 mm ball bearings and 400 µl of 90% methanol in each well were shaken in a Harbil 5-Gallon Mixer (Fluid Management Co., Wheeling, IL) for 3–5 min. Plates were centrifuged at 4000 rpm for 20 min. To bind GSL to Sephadex, 150 µl of supernatant from each well (containing the extracted organic compounds) was transferred to the corresponding well of the 96 well filter plate containing hydrated Sephadex and centrifuged at 1200 rpm for 3 min on top of the 96 deep well discard plate. To wash away all the non-binding organic compounds from the Sephadex, 150 µl of 90% methanol was added to each well and the plate was centrifuged at 1200 rpm for 3 min. To remove excess methanol, two wash steps were conducted by adding 150 µl of $H_2O$ to the plate followed by centrifugation at 1200 rpm for 3 min. To release the GSL from the Sephadex, 10 µl of Sulfatase (Sigma–Alrich) and 100 µl of water were added to each well of the 96 well filter plate then incubated overnight in the dark. The desulfoglucosinolates were then eluted into a 96 well round bottom plate via centrifugation at 1200 rpm for 3 min. For each GSL sample, 50 µl of the 110 µl of extract was injected on an Agilent 1100 HPLC (Agilent, Santa Clara, CA) using a Lichrocart 250–4 RP18e column (Hewlett–Packard, Palo Alto, CA). Individual GSL compounds were detected at 229 nm and separated utilizing the following program with an aqueous acetonitrile gradient: a 6-min gradient from 1.5% to 5.0% (vol/vol) acetonitrile, followed by a 2-min gradient from 5% to 7% (vol/vol)

acetonitrile, a 7-min gradient from 7% to 25% (vol/vol) acetonitrile, a 2-min gradient from 25% to 92% (vol/vol) acetonitrile, 6 min at 92% (vol/vol) acetonitrile, a 1-min gradient from 92% to 1.5% (vol/vol) acetonitrile, and a final 5 min at 1.5% (vol/vol) acetonitrile (*Kliebenstein et al., 2001a*). For each peak, the GSL structure was determined by comparing the retention time and UV absorption spectrum with known standards. The integral under each peak was automatically calculated and this value in mili-absorption units was converted to picamoles/mm$^2$ tissue using response factor slopes determined from purified standards and area of leaf tissue used per sample as measured by ImageJ (*Kliebenstein et al., 2001a*; *Reichelt et al., 2002*).

### Leaf damage measurements in the field

Leaf damage estimates were visually taken in the field at 4 weeks of age, just before tissue collection for GSL extraction. A scale from 0–10 was used to determine amount of pest damage incurred on each plant, with 0 representing no damage and 10 representing complete destruction of the plant (i.e., the plant completely eaten).

### Absolute fitness and relative fitness

Absolute fitness was calculated as total fruit count (TFC) × silique length × survival. TFC was measured as the count of fruits (siliques) + flowers + buds per individual. Silique length was measured in ImageJ from digital images of harvested field plants taken using a Nikon D3100 as follows: each plant was placed flat on a white sheet of paper next to a ruler and pictures were taken using auto focus. After setting the scale in ImageJ using the ruler placed in each image, the segmented line tool was used to draw a line from the pedicle to the tip of the silique. For each plant, eight siliques were measured at random and these values were averaged to get a value for each plant. Survival was scored on a binary (0–1) scale. Plants that germinated, were transplanted into the field and survived to harvest were given a survival score of 1 and plants that germinated and were transplanted but did not survive to harvest were given a score of 0. Individuals that did not germinate or did not survive to transplantation were given an NA. Relative fitness was calculated for each GSL genotype within each environment relative to Col-0. To do this, average absolute fitness of a GSL genotype was divided by the average absolute fitness of Col-0 within a environment. Col-0 was chosen as the reference genotype given that it is the background genotype.

### Statistical analysis methods

All statistical analyses were carried out using the R statistical computing language (*Team, 2014*). The field trial was conducted in a split plot design with each plot nested within treatment by environment. We used a restricted maximum likelihood (REML) approach to fit a linear mixed effects model to the field traits and partition the variation of each among the fixed effects, genotype, environment, treatment and the random factor, plot nested within treatment and environment. There were 17 genotypes, which refers to the GSL genotype in the synthetic laboratory population. There were three environments: Wyoming 2011, Wyoming 2012 and Davis 2012. The two treatments were control and pesticide treated. We had 4 plots per environment (2 in each treatment group) for a total of 12 plots. We used the following formula to fit this model using the lme4 package in R (*Baayen et al., 2008*):

lmer(Trait ~ Genotype*Environment*Treatment + (1|Plot(Treatment:Environment))).

The Anova function from the car package in R was utilized to determine which fixed effects variables significantly altered the mean of each trait (p value <= 0.05) (*Fox and Weisberg, 2011*). We estimated population means (i.e., LSMeans) of each field trait for all genotypes across treatment and environment using the LSMeans function from the doBy package in R (*Højsgaard et al., 2013*). Dunnett's multiple comparison testing was performed on the traits to determine which genotypes had significantly different means than Col-0, our reference genotype using the glht function from the multcomp package in R (*Hothorn et al., 2014*). Additionally, Tukey's multiple comparison was performed on the traits to compare all the genotypes to all the other genotypes for significant differences using the same glht function from the multcomp package in R (*Hothorn et al., 2014*). PCA was conducted using the princomp function from the base package (*Team, 2014*).

## Acknowledgements

We thank Carlos Quiros, Ute Wittstock and Bjarne G Hansen for their generous donations of seed stocks and members of the Kliebenstein and Weinig labs for assistance in the field.

# Additional information

## Competing interests

DJK, Reviewing editor, *eLife.* The other authors declare that no competing interests exist.

## Funding

| Funder | Grant reference | Author |
|---|---|---|
| National Science Foundation (NSF) | DGE 0653984 | Rachel Kerwin |
| National Science Foundation (NSF) | DBI 0820580 | Julie Feusier, Jason Corwin, Catherine Lin, Alise Muok, Brandon Larson, Baohua Li, Bindu Joseph, Daniel Copeland, Daniel J Kliebenstein |
| National Science Foundation (NSF) | MCB 1330337 | Julie Feusier, Jason Corwin, Catherine Lin, Alise Muok, Brandon Larson, Baohua Li, Bindu Joseph, Daniel Copeland, Daniel J Kliebenstein |
| Danish National Research Foundation | DNRF99 | Daniel J Kliebenstein |

The funders had no role in study design, data collection and interpretation, or the decision to submit the work for publication.

## Author contributions

RK, Conception and design, Acquisition of data, Analysis and interpretation of data, Drafting or revising the article, Contributed unpublished essential data or reagents; JF, BL, DC, Acquisition of data, Drafting or revising the article; JC, CW, Analysis and interpretation of data, Drafting or revising the article, Contributed unpublished essential data or reagents; MR, Conception and design, Acquisition of data, Drafting or revising the article; CL, AM, BL, BJ, MF, Acquisition of data, Analysis and interpretation of data, Drafting or revising the article; DJK, Conception and design, Analysis and interpretation of data, Drafting or revising the article

## Author ORCIDs

Daniel J Kliebenstein, http://orcid.org/0000-0001-5759-3175

# Additional files

## Major dataset

The following dataset was generated:

| Author(s) | Year | Dataset title | Dataset ID and/or URL | Database, license, and accessibility information |
|---|---|---|---|---|
| Kerwin RE, Feusier J, Muok A, Larson B, Lin C, Rubin M, Copeland D, Francisco M, Weinig C, Kliebenstein DJ | 2012 | Data from: Natural Genetic Variation in Arabidopsis thaliana Defense Metabolism Genes Modulate Field Fitness | http://dx.doi.org/10.5061/dryad.8qp37/1 | Available at Dryad Digital Repository under a CC0 Public Domain Dedication. |

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
