## [Decision Letter]

Thank you for sending your work entitled “Natural Genetic Variation in *Arabidopsis thaliana* Defense Metabolism Genes Modulate Field Fitness” for consideration at *eLife*. Your article has been favorably evaluated by Ian Baldwin (Senior editor) and three reviewers, one of whom is a member of our Board of Reviewing Editors.

The following individuals responsible for the peer review of your submission have agreed to reveal their identity: Merijn Kant (Reviewing editor) and Michael Turelli (who was consulted by the referees for advice). The other two reviewers remain anonymous.

The Reviewing editor and the other reviewers discussed their comments before we reached this decision, and the Reviewing editor has assembled the following comments to help you prepare a revised submission.

Kerwin and colleagues created a range of glucosinolate mutants in a single genetic background of *Arabidopsis*. These mutants were designed such that in each mutant line distinct branches of the glucosinolate pathway were removed selectively and these predicted phenotypes were validated. Then they composed variable populations by transplanting these mutants together at several field sites during several years to monitor several fitness proxies. The manuscript revolves around the question to which extent specific genes can affect fitness under different environmental conditions differently. The value of this work is that it provides a comprehensive assessment on known genes and fluctuating patterns of herbivory and fitness in nature. Hence, Kerwin and colleagues argue that that fluctuating selection may maintain polymorphisms in glucosinolate biosynthesis genes in natural populations.

The authors extend beyond their data after reasoning that these different environments basically reflect environmental fluctuations and they use their data to parametrize a model of [95] in order to find indications that fluctuating selection could be responsible for maintaining allelic variation in natural *Arabidopsis* populations and their analysis suggests that it does. In addition, they collected 144 *Arabidopsis* accessions to assess if the signature of fluctuating selection can be seen among these accessions as well and this analysis shows non-random variation of glucosinolate haplotypes suggesting that fluctuating selection may maintain genetic variation here as well.

We all agreed that the manuscript reads very well and that the data set is extremely interesting and comprehensive. However, we also have significant concerns with respect to the usage of the model of Turelli and Barton (subsection “Fluctuating selection estimates”) and the interpretation of the analysis performed on the 144 accessions (in the subsection “Non-random variation of GSL loci among field collected accessions”).

1) Firstly, the model of Turelli and Barton cannot be used for this type of data. During our discussion we consulted Michael Turelli directly and together we reached the following conclusion: although these data certainly warrant a discussion on fluctuating selection to play a role here, there is no simple formula to provide the appropriate polymorphism condition in this case, to the best of our knowledge. The detailed comments of Turelli are below and can form an excellent basis for a revised discussion on known genes and fluctuations in herbivory and fitness in nature (we were missing [80], 337:1081 in Science, which during the discussion came up as one of the few, if not only, other examples). Furthermore we suggest you make an estimate of the genetic correlation between environments for levels of herbivore resistance among these genotypes. This will be informative, and should not be difficult to calculate from the existing data and will bring depth into the Discussion. Please take this discussion into account that glucosinolates can have effect on fitness also via other effects than herbivore damage alone.

2) Secondly, we all agreed that the interpretation of the analysis on the 144 field collected accessions is too opportunistic. Non-random variation of glucosinolate loci among natural accessions could be caused by fluctuating selection but there are equally valid alternatives (not mutually exclusive) such as drift, gene flow or historical population structure etc. Now the conclusions worded in the subsection headed “Non-random variation of GSL loci among field collected accessions” are much too selectively biased towards the first. We strongly suggest you rephrase the interpretation thoroughly doing just the alternatives or, and this may be the better alternative, to remove this analysis from your study altogether.

We hope that this letter, the minor comments of the referees and the report of Dr. Turelli will help you to revise your manuscript for *eLife*.

Please find below detailed comments of Michael Turelli in response to the manuscript and the discussion among the referees:

“This is indeed deep water. Given the technical nature of the relevant theory it should be no surprise that both the authors and the reviewers make incorrect assertions, but both make important points.

First, the relevance of the [95] conditions to maintenance of variation involving genotypes created from “mutants [involving] two or three mutated alleles (different loci) [brought] together to get the desired phenotype.” The [95] conditions involve the effects of alleles at individual loci. Hence, one cannot invoke conditions for multilocus genotypes. As I understand this is a key criticism of the reviewers (“the locus effects reported here are confounded by the effects of other loci that are in disequilibrium in this experimental population”). This criticism is correct, but see below.

Second, the relevance of the [95] conditions to *any* genetic variants in *Arabidopsis thaliana*, which is predominantly selfing. [95] explicitly assume random mating. If *Arabidopsis thaliana* were *completely* selfing, which I believe is reasonable as a first approximation, populations would be effectively composed of competing clonal genotypes (the referees mention here that *Arabidopsis* is less clonal than indicated by Turelli, otherwise genome-wide association studies would be impossible). In this case, the distinction between variants at one locus vs multiple loci is irrelevant, as is the distinction between diploidy and haploidy. Hence, the diploid-random-mating conditions provided by [95] are irrelevant to the maintenance of variation. In general, temporal fluctuations alone *cannot* maintain variation for a haploid, the genotype with the highest geometric mean fitness will prevail (this idea goes back to [24], cited in [97], Evolution 55:1283-1298, which explicitly deals with the maintenance of a famous flower-color polymorphism). For such populations, one must invoke either spatially varying fitness or temporal variation with a seed bank. The exact conditions for the maintenance of variation will depend on the nature of gene flow between patches with alternative selection regimes and the extent to which the seed bank creates overlapping generations. These issues are discussed by [97].

The conditions provided by [95] cannot be directly applied to the system of Kliebenstein system. However, showing that different genotypes are favored under different conditions does indeed suggest that fluctuating selection (in time and/or space) may contribute to the maintenance of this polymorphism. The exact mathematical conditions relevant to this system have probably not been worked out. However, assuming near-complete selfing, maintenance of variation will require overlapping generations via a seed bank (I'm not sure if this is relevant to *A. thaliana*) and/or spatial variation with gene flow. The exact conditions will be subtle and depend on biological details that are surely not known.”

Reviewer #2 minor comments:

1) Abstract: “fitness effects were significant in each environment but the pattern fluctuated such that highly fit alleles in one year displayed lower fitness in another.” This experiment doesn't compare alleles, it compares multilocus genotypes.

2) In the subsection headed “Environment interacts with GSL genotype to impact leaf damage”: “no particular GSL genotype showed a consistent maximal or minimal level of leaf damage across the three field trials”. This conclusion that no genotype has consistently highest or lowest damage level obscures the highly significant main effect of genetic differences in damage levels (Table 3).

3) Discussion: “it has been complicated to validate that specific polymorphic loci within a pathway are the actual causative basis of any changes in fitness due to the use of polygenic populations”. Unfortunately, this study shares the same shortcoming, because the chosen combination of multilocus genotypes prevents a clear test for antagonistic pleiotropy at individual loci.

4) Discussion: “we can directly conclude that it is these specific genes and their GSL phenotypes that are determining the differences in fitness in the field”. True, but that is not sufficient to show that fluctuating selection maintains genetic variation at any particular locus.

5) In the subsection “Statistical analysis methods”: Each plot has 10 blocks. Why is there is no block term in the ANOVA?

6) Fitness is normalized with respect to performance of Col-0. This is contrary to the usual definition of relative fitness (with a mean of 1.0). Because of this normalization to Col-0, the reported effects on relative fitness may reflect behavior of Col-0 rather that population means.

7) What does Figure 8 tell us, beyond the main and interaction effects already reported in Table X?

8) In the subsection headed “Pleiotropic Links to GSL Genes”: “while the GSL genes are causing pleiotropic effects, these pleiotropic effects are not driving the observed fitness consequences of the GSL genotypes in our field trials.” A simple pairwise correlation analysis is not sufficient to support this conclusion. More complex ANCOVA-like models might be helpful here, perhaps with principal components of GSLs.

*Reviewer #3 minor comments*:

1) Introduction: this sentence gives the impression that intermediate frequency variants are frequent in the genome. This is misleading, because the majority of variants do have low frequency. It is just that there are several intriguing examples of intermediate frequency alleles. Furthermore, people have tried to relate FRI to fitness and failed in most cases.

2) Also in the Introduction: I wonder why the authors do not mention the work of [28], which does relate underlying genes to fitness.

3) In the subsection “Leaf damage in the field varies across environments”: pesticide application is somehow unfortunate because it artificially decreases herbivory load and thus might explain why the expected effect of GLS variation on fitness via herbivore defense was not observed. This must be better incorporated in the Discussion.

4) In the subsection “Fluctuating selection estimates”: I really appreciate that the authors include now the model of Turelli and Barton, but the model and its predictions should be briefly summarized (for the reader this will come out of the blue!). More explanations are needed in the Methods as well. Did the authors use LSmeans to calculate the parameters? Were LSmeans calculated after correction of error over dispersion?

5) In the subsection “Non-random variation of GSL loci among field collected accessions”: this formulation is somehow strange. The words “population structure” should be mentioned so that the right bell rings for the reader.

6) In the same subsection: the authors should be clearer here that this analysis cannot exclude the possibility that the skewed haplotype distribution can result from population structure only.

7) Discussion: this aspect of the Discussion is still too uncritical for me. The authors should not forget that their findings are not expected and may be driven by processes that were not supposed to play a role. What about insertion effects? Inserted transgenes can disrupt other genes and may result in fitness effects that are unrelated to GLS function. EMS lines initially contain thousands of mutations. They are generally removed by multiple generation of backcrossing to the wild type. Crossing adds some additional backcrossing, is it possible that lines differ in the number of linked mutations? Finally, transgenes are not always stably expressed and could be silenced in one or the other generation, especially if several tDNA insertion lines are coupled. Such problems are well known to (good) molecular biologists but often overseen by ecologists. I believe it is important to critically assess the possibility that the manipulated lines may not be doing exactly what they are supposed to do.

[Editors’ note: an earlier version of this manuscript was also reviewed. The previous decision letter after peer review is shown below.]

Thank you for choosing to send your work entitled “Field Evidence that Fluctuating Selection Can Maintain Natural Genetic Variation in *Arabidopsis thaliana* Defense” for consideration at *eLife*. Your full submission has been evaluated by Ian Baldwin (Senior editor) and three peer reviewers, one of whom, Merijn Kant, is a member of our Board of Reviewing Editors, and the decision was reached after discussions between the reviewers.

The referees appreciated the impressive field experiment and are convinced it has delivered a highly valuable data set that deserves publication. However, the current manuscript needs too much work to warrant a revision for *eLife*. The referees were predominantly concerned about three issues: (1) the framing of the story around the fluctuating selection-theme inferred from the genotype distribution of 144 *Arabidopsis* accessions; (2) the statistical analyses of the field experiment data and (3) on how the fitness proxies came about. I will briefly summarize the main comments which you will also find in more detail in the full referee reports:

1) The referees feel that “fluctuating selection/ bet hedging” theme should only (at best) appear in the Discussion and certainly not be a central theme. Much of this has to do with the analysis presented in Figure 2 using the 144 *Arabidopsis* accessions for drawing conclusions on natural selection as a determinant of their genotype distribution.

2) The ANCOVA analysis using the 144 accessions and its interpretation is troublesome, and maybe should be removed altogether. Several of the remaining statistical analyses (Figures 4, 5, 6, 7 and 8) should be adjusted with respect to model structure, survival and zeroes and multiple testing.

3) The validity of the fitness-proxy parameters should be better justified.

Reviewer #1:

General assessment:

The field experiment is really very interesting and has delivered a valuable data set. I disagree with several aspects of the analysis pipeline and hence I do not see sufficient support for the main conclusions. The article is not easy to read since phrasing is often vague and imprecise.

My main criticisms:

1) The projection of the experimental data onto the genotype-distribution of 144 *Arabidopsis* accessions does not work for me:

A) I fail to see why you would expect to see evidence for natural selection within a group of plants (accessions) of which you do not know what the original selection criteria were: for sure these are not ecotypes and it is unclear to which extent they represent the genotypes of their respective original populations (Weigel, 2012, Plant Physiol: 158:2-22). This should at least be discussed in much more detail.

B) The analysis presented in Figure 2 that justifies the field experiment (in the subsection headed “Structured population mimicking natural GSL variation in Arabidopsis”) is not strong since a test for the goodness of fit of the overall observed vs expected relative frequency distributions is missing (e.g. chi square test for goodness of fit).

C) How does the analysis of the Figure 2 data exclude non-selective processes as an alternative for random assortment (I think neither of these are mutually exclusive).

D) You write “field studies confirm lab results” (at the end of the subsection “GSL genetic variation controls GSL profile in the field”), but you do not provide a direct comparative analysis, just a 'visual' interpretation of the data.

E) In the subsection “Empirical fluctuating measures of selection in the field predict standing variation in GSL genes”, you conclude that “selection likely played a role”, but you do not explain where in the figures we can see this. Looking at the three plots (Figure 9), I only see that the correlation between frequency and your relative fitness-proxy is weak (probably due to UWY2012).

2) You used a “deterministic” approach to genotype the 144 accessions i.e. on the basis of their “GSL profiles” but a validation of this approach is missing:

A) Throughout the manuscript is remains unclear what is meant with “profile” and how these were evaluated (e.g. which grouping criteria/procedures/assignment to a genotype). Hence its validity cannot be assessed.

B) In the subsection “GSL genetic variation controls GSL profile in the field”, it is unclear to which extent actual profile information was used for the downstream analyses or when only the information on the total (aliphatic) glucosinolates was used (e.g. see Figures 4 and 5). Figure 4 appears to represent these “profiles” but a statistical evaluation is not provided.

C) Why was this indirect approach for genotyping preferred over a direct (DNA-based) approach?

3) The factors of the statistical analyses are often unclear:

A) The factor “location” is misleading: it should be “environment” since the different locations were used at different moments in time. Why do you refer to these differences as 'fluctuation' i.e. how do you know they are not caused solely by differences in starting conditions? Now you infer these differences from Figure 5 but the patterns across the three panels should be statistically evaluated to decide to which extend they differ.

B) The statistical “interaction” is misinterpreted and dance around its meaning. Significant interactions indicate that their simultaneous effects are not additive i.e. either the combined effect is greater (synergistic) or smaller (antagonistic) than expected (additive) effect. Pinpointing what they mean is sometimes virtually impossible and requires post hoc statistics.

C) The ANCOVA procedure is not explained anywhere.

4) Your fitness proxies, and especially how they were normalized, need validation. In “GSL variation impacts fitness in the field”, you describe a normalization procedure (which assumes a linear relationship between silique length and fecundity) which struck me as highly arbitrary. This procedure needs references or a solid validation.

Reviewer #2:

This manuscript describes the performance in the field of 20 lines differing only in 8 loci controlling glucosinolate synthesis. The author used 20 lines combining one or more variant at these loci. They planted them in a random block design, in 40 replicates, in two consecutive years in WY and in 2012 in UC Davis. In each experimental block, the level of herbivory was manipulated with the help of pesticides.

They monitored the glucosinolate profile of these lines in the field and observed that these profiles broadly match with those observed in the lab. They further quantified leaf damage and demonstrated that the genetic differences between these lines alter leaf damage levels. Finally, the authors present evidence that the fitness of individual lines differ within and between sites/years.

In order to address the relevance of the specific observations they made at these two sites, the authors also used known glucosinolate levels to predict the relative occurrence of the glucosinolate profiles in a set of 144 natural accessions. They report that the lines that have the highest fitness in one of their field site/year tend to be the ones whose genotype is overrepresented in the natural population.

This work is unique: the system is very well known, the genetic differences between the lines alter glucosinolate production in the field and the match between observed frequencies and fitness is very exciting. To my knowledge, this is the first time that the fitness consequences of an extent molecular system is studied in the field. This work demonstrates that glucosinolate production is important for fitness. I find equally interesting that this fitness relevance cannot be explained by insect damage. This shows that the ecological function of glucosinolate is not as straightforward as one may think. Finally, the authors document the complexity of field studies in *A. thaliana* where variation across year and location dramatically alter fitness levels.

This work is excellent and I strongly recommend it for publication in *eLife*.

Reviewer #3:

This manuscript examines herbivory and fitness in field sites in the USA for a series of transgenic glucosinolate genotypes, and compares these local fitness estimates to the frequency of genotypes in 144 European *Arabidopsis* accessions. The authors conclude that the relationship between field fitness and observed glucosinolate genotype frequencies is due to fluctuating selection pressures that maintain genetic variation in nature. The questions are of great ecological and evolutionary significance, but these conclusions are compromised by statistical and evolutionary shortcomings.

Statistics:

1) This is a split plot design (for insecticide treatment), not a randomized complete blocks design. Correction of this analysis will certainly affect inferences related to the insecticide treatment, and may alter other parts of the model.

2) Inclusion of individuals with zero fitness is not compatible with the distributional assumptions of this ANOVA. This should be visible in the residuals, although no statement is made regarding such verification of statistical assumptions. Such zero-inflated data pose a difficult problem in such analyses, which typically alter levels of statistical significance and often cause spurious rejection of the null hypothesis.

3) Controls for multiple statistical tests are needed at several points in this manuscript. Examples include tests for non-random distribution of multilocus GSL genotypes (Table 2) comparison of treatment effects, and variation of GSL among sites (Figure 5).

Evolution and Genetics:

4) The evolutionary significance of this study is justified as a test whether fluctuating selection maintains genetic variation within and among populations. However, these experiments do not estimate herbivory or fitness for the individual GSL loci, and they do not show change in rank fitness at putatively selected loci, which is a necessary condition for balancing selection to maintain non-neutral genetic polymorphism.

5) Even if the patterns in Figure 8 were due to selection, they might be attributable to non-equilibrium directional selection in Eurasia, rather than to historical balancing (fluctuating) selection. Consequently, this analysis cannot prove that such patterns are due to fluctuating pressures maintaining standing natural variation within a species.

6) The observation that multi-locus genetic variation controlling aliphatic GSL appears to be non-randomly distributed among the natural accessions is interpreted as evidence for natural selection. However, this may also result from population structure, nonrandom geographic sampling, finite population size, or failure to correct for multiple tests.

7) “We observed significant variation in silique lengths across genotypes” How do GSL polymorphisms alter silique length? And, how do these effects differ among pesticide treatments? Alternatively, rather than the effects of GSL polymorphisms, variation in silique length (and fitness) may be due to position effects of the transgene inserts, or untagged Agrobacterium hits, or linked mutations not eliminated following EMS.

8) Several points weaken the proposed role of herbivory in shaping the observed patterns in GSL polymorphisms:

8A) In Figure 6 we see that the ranking of herbivore damage (without insecticide) decreases from UWY2012 > UCD2012 > UWY2011. However, the correlations in Figure 12 show the opposite pattern: UWY2012 < UCD2012 < UWY2011. This seems incompatible with the conclusion that the herbivory differences in the field reflect natural selection by herbivores in Eurasian Arabidopsis over thousands of generations.

8B) Similarly, the authors note that “the positive correlation between observed genotype frequency and fitness disappeared in the high herbivory WY2012 field trial.” This suggests that these results may not be due to herbivory.

8C) To test for herbivory-mediated effects, one could ask whether the ANCOVA is significant if only the no-pesticide treatment is analyzed. Or, what happens to the patterns in Figure 8 if the change in fitness between insecticide treatments is used as the response variable for the ANCOVA?

---

## [Author Response]

*We all agreed that the manuscript reads very well and that the data set is extremely interesting and comprehensive. However, we also have significant concerns with respect to the usage of the model of Turelli and Barton (subsection “Fluctuating selection estimates”) and the interpretation of the analysis performed on the 144 accessions (in the subsection “Non-random variation of GSL loci among field collected accessions”)*.

*1) Firstly, the model of Turelli and Barton cannot be used for this type of data. During our discussion we consulted Michael Turelli directly and together we reached the following conclusion: although these data certainly warrant a discussion on fluctuating selection to play a role here, there is no simple formula to provide the appropriate polymorphism condition in this case, to the best of our knowledge. The detailed comments of Turelli are below and* can *form an excellent basis for a revised discussion on known genes and fluctuations in herbivory and fitness in nature (we were missing*
[80]*, 337:1081 in Science, which during the discussion came up as one of the few, if not only, other examples). Furthermore we suggest you make an estimate of the genetic correlation between environments for levels of herbivore resistance among these genotypes. This will be informative, and should not be difficult to calculate from the existing data and will bring depth into the Discussion. Please take this discussion into account that glucosinolates* can *have effect on fitness also* via *other effects than herbivore damage alone*.

We have removed the sections using the Turelli approach and we have included a new section in the Discussion commenting on what would be further required to allow for fluctuating selection to maintain diversity in *Arabidopsis* (i.e. seed bank, etc) and provided citational support for these requirements in *Arabidopsis* within the field. Finally, we have concluded this section with a comment on the dramatic need for further field trials and natural history studies to test for this potential in the field.

We apologize for having not included the Prasad citation as we had been focusing on single gene manipulations. We have now included this citation.

We have now included the genetic correlation of herbivory across the three environments that shows that there is no correlation which agrees with the concept of fluctuating herbivory pressures. We would also like to note that we have an entire paragraph in the Discussion (third paragraph) on glucosinolates altering fitness via their effects on other processes that may influence resistance to unmeasured abiotic fluctuations. We hope that this is sufficient.

*2) Secondly, we all agreed that the interpretation of the analysis on the 144 field collected accessions is too opportunistic. Non-random variation of glucosinolate loci among natural accessions could be caused by fluctuating selection but there are equally valid alternatives (not mutually exclusive) such as drift, gene flow or historical population structure etc. Now the conclusions worded in the subsection headed “Non-random variation of GSL loci among field collected accessions” are much too selectively biased towards the first. We strongly suggest you rephrase the interpretation thoroughly doing just the alternatives or, and this may be the better alternative, to remove this analysis from your study altogether*.

We would like to keep this analysis as we feel it nicely connects to the empirical data and provides an alternative explanation for the distribution of variation in the accessions. We have added the following material to this section in the hopes of properly caveating the data and saying that future work needs to be done to clarify any of the hypotheses about the data: “It is similarly possible that this non-random variation is caused by non-selective processes like migration, population structure and local bottlenecks. Significant future efforts will be required to test the extent to which this non-random variation is caused by neutral demographic processes vs potential fluctuating selection.” We feel that we would be remise if we did not include this data and hope that this more detailed caveating of all possible explanations for interpretation helps. If it is still felt that this is too much we will remove it.

*We hope that this letter, the minor comments of the referees and the report of Dr. Turelli will help you to revise your manuscript for* eLife*.*

*Please find below detailed comments of Michael Turelli in response to the manuscript and the discussion among the referees*:

“*This is indeed deep water. Given the technical nature of the relevant theory it should be no surprise that both the authors and the reviewers make incorrect assertions, but both make important points*.

*First, the relevance of the*
[95]
*conditions to maintenance of variation involving genotypes created from* “*mutants [involving] two or three mutated alleles (different loci) [brought] together to get the desired phenotype.*” *The*
[95]
*conditions involve the effects of alleles at individual loci. Hence, one cannot invoke conditions for multilocus genotypes. As I understand this is a key criticism of the reviewers (*“*the locus effects reported here are confounded by the effects of other loci that are in disequilibrium in this experimental population*”*). This criticism is correct, but see below*.

*Second, the relevance of the*
[95]
*conditions to any genetic variants in* Arabidopsis thaliana*, which is predominantly selfing.*
[95]
*explicitly assume random mating.* If Arabidopsis thaliana *were completely selfing, which I believe is reasonable as a first approximation, populations would be effectively composed of competing clonal genotypes (the referees mention here that* Arabidopsis *is less clonal than indicated by Turelli, otherwise genome-wide association studies would be impossible). In this case, the distinction between variants at one locus versus multiple loci is irrelevant, as is the distinction between diploidy and haploidy. Hence, the diploid-random-mating conditions provided by*
[95]
*are irrelevant to the maintenance of variation. In general, temporal fluctuations alone cannot maintain variation for a haploid, the genotype with the highest geometric mean fitness will prevail (this idea goes back to*
[24]*, cited in*
[97]*, Evolution 55:1283-1298, which explicitly deals with the maintenance of a famous flower-color polymorphism). For such populations, one must invoke either spatially varying fitness or temporal variation with a seed bank. The exact conditions for the maintenance of variation will depend on the nature of gene flow between patches with alternative selection regimes and the extent to which the seed bank creates overlapping generations. These issues are discussed by*
[97].

*The conditions provided by*
[95]
*cannot be directly applied to the system of Kliebenstein system. However, showing that different genotypes are favored under different conditions does indeed suggest that fluctuating selection (in time and/or space) may contribute to the maintenance of this polymorphism. The exact mathematical conditions relevant to this system have probably not been worked out. However, assuming near-complete selfing, maintenance of variation will require overlapping generations* via *a seed bank (I'm not sure if this is relevant to* A. thaliana*) and/or spatial variation with gene flow. The exact conditions will be subtle and depend on biological details that are surely not known*.*”*

We have added in a point in the Discussion section, stating that for fluctuating selection to maintain diversity in *Arabidopsis* would require either a seed bank and/or spatial separation of selection. We then cited papers providing evidence of both and commented on the need for more natural history to provide detailed parameters to these to allow for more detailed models to be built for this situation. We hope that this helps.

Reviewer #2 minor comments:

*1) Abstract:* “*fitness effects were significant in each environment but the pattern fluctuated such that highly fit alleles in one year displayed lower fitness in another.*” *This experiment doesn't compare alleles, it compares multilocus genotypes*.

We agree that this experiment compares different multilocus genotypes rather than alleles. The text has been changed and “alleles” has been changed to “genotypes”.

*2) In the subsection headed “Environment interacts with GSL genotype to impact leaf damage”:* “*no particular GSL genotype showed a consistent maximal or minimal level of leaf damage across the three field trials*”*. This conclusion that no genotype has consistently highest or lowest damage level obscures the highly significant main effect of genetic differences in damage levels (*Table 3*)*.

We agree that genotype is significant but the genotype x environment term is more significant and contains more than two times the variance even though linear models are known to overweight the main effect terms at the cost of the interaction terms. This is in agreement with the visual analysis in the figures where the maximal and minimally fit genotypes are inconsistent. We think the middle is where the main effect variance is coming from. Thus, we prefer keeping our original conclusion.

*3) Discussion:* “*it has been complicated to validate that specific polymorphic loci within a pathway are the actual causative basis of any changes in fitness due to the use of polygenic populations*”*. Unfortunately, this study shares the same shortcoming, because the chosen combination of multilocus genotypes prevents a clear test for antagonistic pleiotropy at individual loci*.

We feel that this shortcoming is by no means the same scale as the existing polygenic populations that are segregating for thousands of genes some in extreme linkage (Kroymann and Mitchell-Olds, 2005, Nature 435, 95-98). In our case, we have a mere handful of discrete gene manipulations all within a single pathway that allows us to focus our interpretation in a way that was not previously possible. We have commented on how it would be optimal to have the full 256 line matrix containing all combinations of alleles between all loci to fully interrogate the effects of all loci in all possible backgrounds (please see the Discussion).

*4) Discussion:* “*we* can *directly conclude that it is these specific genes and their GSL phenotypes that are determining the differences in fitness in the field*”*. True, but that is not sufficient to show that fluctuating selection maintains genetic variation at any particular locus*.

We tried very carefully to re-edit the manuscript to ensure that we aren’t making specific allele claims but instead talking in general about the group of loci that we manipulated (i.e. genotypes). We hope that our significant caveats throughout this paper are sufficient to convey to the reader that we are simply providing evidence supporting a role of fluctuating selection in maintaining variance and are not arguing that we have proven this as the sole mechanism at play. As stated earlier in the letter from the editor, there are almost no studies providing similar evidence and we hope that by leaving the paper as written with it being a hypothesis that we can stimulate further research.

*5) In the subsection “Statistical analysis methods”: Each plot has 10 blocks. Why is there is no block term in the ANOVA*?

The blocks were subtended to the plot term as suggested in the previous round of review for the split-plot analysis and as such we did not investigate the block term.

*6) Fitness is normalized with respect to performance of Col-0. This is contrary to the usual definition of relative fitness (with a mean of 1.0). Because of this normalization to Col-0, the reported effects on relative fitness may reflect behavior of Col-0 rather that population means*.

We have adjusted the normalization to make relative fitness to reflect the population mean. Figures and tables have been adjusted accordingly.

The results of the mixed model analysis do not change depending on whether fitness is normalized relative to Col-0 or the population mean. However, we understand that it might be more common to see relative fitness calculated based on the population mean. We have changed accordingly and apologize for any confusion it may have caused.

*7) What does*
Figure 8
*tell us, beyond the main and interaction effects already reported in*
*Table X*?

This is a graphical representation of the data presented in the table along with the hierarchical clustering of the genotypes. Some readers approach data more graphically while others prefer the tabular form and as such we felt it was best to include both presentations.

*8) In the subsection headed “Pleiotropic Links to GSL Genes”:* “*while the GSL genes are causing pleiotropic effects, these pleiotropic effects are not driving the observed fitness consequences of the GSL genotypes in our field trials.*” *A simple pairwise correlation analysis is not sufficient to support this conclusion. More complex ANCOVA-like models might be helpful here, perhaps with principal components of GSLs*.

We feel that a simple pairwise correlation is sufficient to state that pleiotropic effects of flowering time are not sufficient to explain the entirety of the data. If flowering time was the sole driver, then there should have been a simple pairwise correlation. A more involved ANCOVA or path analysis would be required to delve even further into cause and effect but we feel that this is beyond the scope of this manuscript which is solely working to show that there can be fluctuating selection without giving the specific mechanism which as we stated in the discussion may be beyond our current measurements and require further work.

Reviewer #3 minor comments:

*1) Introduction: this sentence gives the impression that intermediate frequency variants are frequent in the genome. This is misleading, because the majority of variants do have low frequency. It is just that there are several intriguing examples of intermediate frequency alleles. Furthermore, people have tried to relate FRI to fitness and failed in most cases*.

We have changed this section completely to better relate the frequency of alleles topic better. We hope that this works.

*2) Also in the Introduction: I wonder why the authors do not mention the work of*
[28]
*which does relate underlying genes to fitness*.

Thank you for this suggestion. We have added this citation and another study on local adaptation using GWA studies (Platt, 2010 and Li, 2014). We apologize for not including these previously as we were focused on citing studies using specific single locus manipulations in the field.

*3) In the subsection “Leaf damage in the field varies across environments”: pesticide application is somehow unfortunate because it artificially decreases herbivory load and thus might explain why the expected effect of GLS variation on fitness* via *herbivore defense was not observed. This must be better incorporated in the Discussion*.

We had intentionally included the pesticide treatment to enable us to test the effect of herbivory load upon the fitness effects. Unfortunately the fluctuating environments lead to most years having lower than expected herbivory load in these experiments causing the treatment to have minimal impact on the results. We have incorporated a new sentence in this section of the manuscript stating that another explanation for this lack of linkage is that the experiment was too small and needs to be larger. We hope that this is sufficient. We would prefer to not focus too much effort on the lack of an observation, as this is in all likelihood an issue of power.

*4) In the subsection “Fluctuating selection estimates”: I really appreciate that the authors include now the model of Turelli and Barton, but the model and its predictions should be briefly summarized (for the reader this will come out of the blue!). More explanations are needed in the Methods as well. Did the authors use LSmeans to calculate the parameters? Were LSmeans calculated after correction of error over dispersion*?

As suggested by the reviewing editor, we have removed the Turelli and Barton equation work.

*5) In the subsection “Non-random variation of GSL loci among field collected accessions”: this formulation is somehow strange. The words* “*population structure*” *should be mentioned so that the right bell rings for the reader*.

We have better stated the set of demographic/non-selective processes that could have generated a non-random population structure in this section.

*6) In the same subsection: the authors should be clearer here that this analysis cannot exclude the possibility that the skewed haplotype distribution* can *result from population structure only*.

Please see the comment to the editor at the beginning on how we have more fully caveated this section and said that future work needs to be done in the field to assess the relative contribution of selective and non-selective processes to this distribution.

*7) Discussion: this aspect of the Discussion is still too uncritical for me. The authors should not forget that their findings are not expected and may be driven by processes that were not supposed to play a role. What about insertion effects? Inserted transgenes* can *disrupt other genes and may result in fitness effects that are unrelated to GLS function. EMS lines initially contain thousands of mutations. They are generally removed by multiple generation of backcrossing to the wild type. Crossing adds some additional backcrossing, is it possible that lines differ in the number of linked mutations? Finally, transgenes are not always stably expressed and could be silenced in one or the other generation, especially if several tDNA insertion lines are coupled. Such problems are well known to (good) molecular biologists but often overseen by ecologists. I believe it is important to critically assess the possibility that the manipulated lines may not be doing exactly what they are supposed to do*.

We have added in the following material to the Discussion: “We should also note that even with all of our efforts to clean up the respective backgrounds and validate that the mutant phenotypes are similar to the segregating natural genotypes, it remains possible that some of the observed effects are caused by unexpected changes in the lines”.

We hope that with the significant efforts at cleaning up the material as described in the Materials and methods that this is sufficient. We would like to note that we can rule out T-DNA silencing as we phenotyped every plant in the analysis using HPLC so that we can be sure that all insertions were generating the appropriate biochemical phenotype in each and every line and as such the insertions were not silenced.

*8) Finally, the manuscript relies on quantitative analyses of variation. One should not dismiss the possibility that means were not correctly estimated, especially with over dispersed measurement of fitness. There is no such thing as a perfect statistical analysis*.

We hope that the manuscript is now sufficiently caveated throughout to represent the potential pitfalls. We would like to note that as in the previous round of reviews that we had gone through the data to limit any effects of over-dispersion on the mean estimation.

[Editors’ note: the author responses to the previous round of peer review follow.]

*1) The referees feel that* “*fluctuating selection/bet hedging*” *theme should only (at best) appear in the Discussion and certainly not be a central theme. Much of this has to do with the analysis presented in*
Figure 2
*using the 144* Arabidopsis *accessions for drawing conclusions on natural selection as a determinant of their genotype distribution*.

We have extensively rewritten the entire manuscript to focus on validating fitness effects using specific genetic manipulations and field trials. As per the suggestion by the reviewers, we have introduced a new section where we utilize the [95] equations to show that our empirical values for fitness on the GSL genotypes in our population fit within the range of parameters established as necessary for fluctuating selection to stabilize genetic variation as established in this paper (Turelli, 2004). We feel that this combination of rewriting in addition to testing the Turelli and Barton parameters allows the paper to show that these genes/loci affect fitness in the field and may be under fluctuating selection. We have moved Figure 2 to Figure 9 as it is now used to indicate that the natural accessions may support the Turelli and Barton model.

*2) The ANCOVA analysis using the 144 accessions and its interpretation is troublesome and maybe should be removed altogether*.

As suggested by the reviewer, we have removed the ANCOVA analysis.

*Several of the remaining statistical analyses (*Figures 4, 5, 6, 7 and 8*) should be adjusted*.

We have redone all the statistics using the split-plot model structure and previous Figure 8 was removed as it is no longer appropriate. We hope that this rectifies this concern.

*Survival and zeroes*:

We have run all the statistics using absolute fitness with and without survivorship to show that the models have the same result and that the non-survivors are not driving the observed link between genotype and fitness. We hope that this rectifies any concerns.

*Multiple testing*:

We have removed previous Figure 8 as it no longer fits and all other multiple comparisons were done using either Dunnett’s or Tukey’s post-hoc tests that adjusts for multiple testing. We hope that this rectifies this concern.

*3) The validity of the fitness-proxy parameters should be better justified*.

We have updated this section extensively to justify the inclusion of silique length in our measure of absolute fitness. The basis of the argument is that because seed size is similar amongst all GSL genotypes and silique count is an approximation of the potential for seed production, that including silique length, which in *Arabidopsis* is linear to seed size, we have a better approximation of seed production in these individuals. The linearity argument arises from the fact that *Arabidopsis* siliques contain only two lines of seeds per silique and as such a silique is really a two-dimensional fruit with regards to seed number. Therefore, we calculated absolute fitness as total fruit count (TFC) x silique length with and without survivorship (Table 5 and Figure 7 and Figure 8). In addition, we conducted all the statistics using TFC alone with and without survivorship and got similar results ([Supplementary-material SD2-data] and Figure 7—figure supplement 1).

Reviewer #1:

*General assessment*:

*The field experiment is really very interesting and has delivered a valuable data set. I disagree with several aspects of the analysis pipeline and hence I do not see sufficient support for the main conclusions. The article is not easy to read since phrasing is often vague and imprecise*.

*My main criticisms*:

*1) The projection of the experimental data onto the genotype-distribution of 144* Arabidopsis *accessions does not work for me*:

*A) I fail to see why you would expect to see evidence for natural selection within a group of plants (accessions) of which you do not know what the original selection criteria were: for sure these are not ecotypes and it is unclear to which extent they represent the genotypes of their respective original populations (Weigel, 2012, Plant Physiol: 158:2-22). This should at least be discussed in much more detail*.

We have worked to address this concern in several ways. The first is that we have included a new figure showing geographic origin of the accessions (Figure 2) that shows that we are using a globally distributed collection. Secondly, we have moved this analysis of the 144 accessions and corresponding figure to the end of the Results (Figure 9) as it was only meant to be supportive of our conclusions and it became obvious from all the reviewers comments that having it near the beginning was giving it dramatically more weight than expected. We hope that this helps to clarify the meaning of this analysis by placing it at the end of the manuscript. We would also like to note that we did not use the term ecotype but instead accession for the very reason mentioned by the reviewer.

*B) The analysis presented in*
Figure 2
*that justifies the field experiment (in the subsection headed “Structured population mimicking natural GSL variation in* Arabidopsis*”) is not strong since a test for the goodness of fit of the overall observed versus expected relative frequency distributions is missing (e.g. chi square test for goodness of fit)*.

We have now given the goodness of fit for the overall model as requested, which was highly significant (p value <<< 0.001).

*C) How does the analysis of the*
Figure 2
*data exclude non-selective processes as an alternative for random assortment (I think neither of these are mutually exclusive)*.

As stated above, we have moved this figure to the end of the manuscript and made it clearer that this is not proof of selective processes but does support our conclusions. We also discuss that more field trials over successive years as well as more extensive accession sampling would be necessary to fully validate a fluctuating selection model.

*D) You write* “*field studies confirm lab results*” *(at the end of the subsection “GSL genetic variation controls GSL profile in the field”), but you do not provide a direct comparative analysis, just a 'visual' interpretation of the data*.

We have now included a direct comparative analysis using PCA and correlation to show that the GSL profiles of the lab and field grown plants are highly comparable.

*E) In the subsection “Empirical fluctuating measures of selection in the field predict standing variation in GSL genes”, you conclude that* “*selection likely played a role*”*, but you do not explain where in the figures we* can *see this. Looking at the three plots (*Figure 9*), I only see that the correlation between frequency and your relative fitness-proxy is weak (probably due to UWY2012)*.

We agree that the Figure 9 and the corresponding ANCOVA does not strongly support our conclusions and has been removed from the manuscript. It was replaced with calculations of the [95] fluctuating selection model parameters. We hope that this has addressed the concerns.

*2) You used a* “*deterministic*” *approach to genotype the 144 accessions i.e. on the basis of their* “*GSL profiles*” *but a validation of this approach is missing*:

*A) Throughout the manuscript is remains unclear what is meant with* “*profile*” *and how these were evaluated (e.g. which grouping criteria/procedures/assignment to a genotype). Hence its validity cannot be assessed*.

We have worked to better explain that profile is the mixture of GSLs within a genotype and their relative abundance. To support this description, we have included new figure supplements to Figure 1 that provide representative HPLC trace outputs for each GSL genotype which is a direct visual of what the GSL profile looks like to better illustrate this concept within this system. The way profile is defined has been expanded in the text. This was clearly an oversight and we apologize for the confusion. We have explicitly provided the rules for calling these profiles and their associated haplotypes in [Supplementary-material SD7-data].

*B) In the subsection “GSL genetic variation controls GSL profile in the field”, it is unclear to which extent actual profile information was used for the downstream analyses or when only the information on the total (aliphatic) glucosinolates was used (e.g. see*
Figures 4 and 5*).*
Figure 4
*appears to represent these* “*profiles*” *but a statistical evaluation is not provided*.

We have included statistical analysis on the profile figures as requested and provided visuals on the profiles in Figure 1. Additionally we have worked to make it explicit when we are talking about profile vs the total of aliphatic glucosinolates.

*C) Why was this indirect approach for genotyping preferred over a direct (DNA-based) approach*?

As is true for other naturally variable loci in *Arabidopsis*, such as flowering time, there are independently evolved alleles in the GSL pathway that generate the same phenotype but appear as independent genotypes. Thus, we prefer to define the accessions by their functionality at each locus which better reflects the GSL profile differences observed. Additionally, the GSL loci have local rearrangements that make them nearly impossible to obtain accurate genotypic information using Illumina resequencing approaches and as such we have found that the data in the 1001 databases for these loci is inaccurate when we compare to direct BAC sequencing of the same accession. Thus, the available data is unable to provide accurate genotyping information which is another reason to use the indirect approach at this time.

*3) The factors of the statistical analyses are often unclear*:

*A) The factor* “*location*” *is misleading: it should be* “*environment*” *since the different locations were used at different moments in time. Why do you refer to these differences as 'fluctuation' i.e. how do you know they are not caused solely by differences in starting conditions? Now you infer these differences from*
Figure 5
*but the patterns across the three panels should be statistically evaluated to decide to which extend they differ*.

We have changed the term “location” to “environment” as requested. We worked to minimize the differences in starting conditions by working to make the planting time as close as possible. Similarly, all plants were started in a greenhouse to minimize variation in starting conditions on the seedlings. As with any field trial it is very difficult to ascribe specific causes of fitness differences and we have worked to show that flowering time which should be equally susceptible to starting conditions was not linked to the resulting fitness. Even if the planting data was identical, the starting condition across years could be strikingly different and we interpret that as a fluctuation. We have included statistical analysis within Figure 5 and Table 3 to show the differences.

*B) The statistical* “*interaction*” *is misinterpreted and dance around its meaning. Significant interactions indicate that their simultaneous effects are not additive i.e. either the combined effect is greater (synergistic) or smaller (antagonistic) than expected (additive) effect. Pinpointing what they mean is sometimes virtually impossible and requires post hoc statistics*.

We have included a new set of figures on both herbivory and fitness to show that neither the synergistic nor antagonistic models fit the observed genotype × environment interactions as there are numerous instances of the GSL genotypes crossing, i.e. a genotype’s rank for fitness changing from environment to environment. We hope that this visual analysis and accompanying changes to the statistical model help to show that this fits neither the synergistic or antagonistic model but instead is a better fit to a fluctuating model.

*C) The ANCOVA procedure is not explained anywhere*.

The ANCOVA and the corresponding figure have been removed as per requested by other reviewers.

*4) Your fitness proxies, and especially how they were normalized, need validation. In “GSL variation impacts fitness in the field”, you describe a normalization procedure (which assumes a linear relationship between silique length and fecundity) which struck me as highly arbitrary. This procedure needs references or a solid validation*.

We have built our fecundity procedures on the cited references from Schmitt and colleagues that are inherently built on a linear assumption of silique number and fecundity. At its heart, this makes an implicit assumption that silique number and seed set are directly related. We however found that silique length was genetically determined and as such this implicit assumption may be incorrect. In *Arabidopsis*, the silique is a linear fruit with two files of seeds. We found that seed size was not different amongst genotypes and thus silique length is directly correlated to the number of seeds that can be within that silique. Thus as we had genetic variation in silique length we had to adjust the approximation of Schmitt and colleagues to appropriately reflect this fact. We have worked to make this clearer within the text. In addition, we conducted all the statistics using total fruit count alone with and without survivorship and got similar results as we did for absolute fitness ([Supplementary-material SD2-data] and Figure 7—figure supplement 1).

Reviewer #3:

*Statistics*:

*1) This is a split plot design (for insecticide treatment), not a randomized complete blocks design. Correction of this analysis will certainly affect inferences related to the insecticide treatment, and may alter other parts of the model*.

We have redone the entire analysis with a split plot analysis. All statistics throughout the manuscript have been adjusted to reflect this new analysis. This as suggested by the reviewer lead to a loss of significance of the treatment effect which required us to rework the entire manuscript to remove any inference about the effects of herbivory upon the analysis as there was no difference in the pesticide and control plots. The observation that GSL gene variation and an interaction of GSL gene variation and environment are linked to fitness was still significant.

*2) Inclusion of individuals with zero fitness is not compatible with the distributional assumptions of this ANOVA. This should be visible in the residuals, although no statement is made regarding such verification of statistical assumptions. Such zero-inflated data pose a difficult problem in such analyses, which typically alter levels of statistical significance and often cause spurious rejection of the null hypothesis*.

We have empirically tested if the zero fitness individuals were causing spurious significance by running the model using fitness as calculated with the zeros and without the zero individuals. This is now shown in the manuscript in Table 5 where we show that both models lead to the same interpretation of the data. The residuals are also not significantly affected by the zeros because they are relatively infrequent in comparison to the non-zeros. Throughout the manuscript we have presented analysis using fitness with and without survivorship to allow the reader to specifically compare the two results so that they can confirm our conclusions are not driven by survivorship.

*3) Controls for multiple statistical tests are needed at several points in this manuscript. Examples include tests for non-random distribution of multilocus GSL genotypes (*Table 2*) comparison of treatment effects, and variation of GSL among sites (*Figure 5*)*.

As per the request from Reviewer 1, we now include the whole-model Chi-square goodness of fit for Figure 9 (old Figure 2) and show that the whole model is highly non-significant. The remaining p values in this table are simply to provide indications of which genotypes are more or less deviating within that model. For the other figures, we apologize that we had not noted that we had conducted various post-hoc tests, such as Tukey’s or Dunnett’s to account for multiple correction. This has now been clarified in all instances. Because there was no significant treatment effect, we have removed the comparison from the manuscript.

*Evolution and Genetics*:

*4) The evolutionary significance of this study is justified as a test whether fluctuating selection maintains genetic variation within and among populations. However, these experiments do not estimate herbivory or fitness for the individual GSL loci, and they do not show change in rank fitness at putatively selected loci, which is a necessary condition for balancing selection to maintain non-neutral genetic polymorphism*.

We have included a new figure showing that the ranks of the GSL genotypes are changing across the environments (Figure 7). Additionally, we have added a new section where we utilize the Turelli and Barton model to estimate the per locus *v*i and the population K to show that the empirical values we have found are sufficient to fit the necessary conditions as modeled by Turelli and Barton to allow for fluctuating selection to maintain multi-locus genetic polymorphisms. Additionally, we have reworked the entire manuscript to focus more on the fact that the GSL genes affect field fitness. We hope that this helps to address these concerns as well as makes for a better manuscript as a whole.

*5) Even if the patterns in*
Figure 8
*were due to selection, they might be attributable to non-equilibrium directional selection in Eurasia, rather than to historical balancing (fluctuating) selection. Consequently, this analysis cannot prove that such patterns are due to fluctuating pressures maintaining standing natural variation within a species*.

We agree with the reviewer that proof of how historical selection has occurred is nearly impossible to obtain. Throughout the manuscript we had attempted to state that we have generated data that supports a fluctuating hypothesis but that more extensive field trials and more extensive inter and intra population sampling within the species range are required to further validate this hypothesis. We have worked through the manuscript to ensure that we do not state we have generated proof but instead simply support.

*6) The observation that multi-locus genetic variation controlling aliphatic GSL appears to be non-randomly distributed among the natural accessions is interpreted as evidence for natural selection. However, this may also result from population structure, nonrandom geographic sampling, finite population size, or failure to correct for multiple tests*.

We apologize for the obvious confusion on this section and figure which we had meant to state that the pattern may result from natural selection, population structure, sampling or any other demographic process and then to state that separating between these options needed field tests of specific genetic variants. Instead of having this analysis at the front, we have moved this to the last figure (Figure 9) to indicate that the natural variation in GSL chemotype observed among *Arabidopsis thaliana* accessions agree with the field trial estimates of the Turelli and Barton parameters that support the hypothesis that fluctuating selection can balance multi-locus polygenic genetic variation within the aliphatic GSL pathway.

*7)* “*We observed significant variation in silique lengths across genotypes*” *How do GSL polymorphisms alter silique length? And, how do these effects differ among pesticide treatments? Alternatively, rather than the effects of GSL polymorphisms, variation in silique length (and fitness) may be due to position effects of the transgene inserts, or untagged Agrobacterium hits, or linked mutations not eliminated following EMS*.

We have worked to make it clearer in the text that the genetic backgrounds of our GSL genotype are unlikely to have second site insertions due to the fact that they have been back-crossed multiple times to eliminate unlinked effects. Additionally, the majority of lines are specific insertions within the GSL gene to create the natural knockout allele. Further, in reply to Reviewer 2, we have noted that our previous work has shown that unexpected links to non-GSL phenotypes, such as flowering time, have been validated both with the transgenic and natural populations i.e. we could map circadian timing variation to GSL loci in an *Arabidopsis* RIL populations and then show the effect using the transgenic validation lines. This was with both enzymatic loci and transcription factor loci showing that this is not a second site effect, but instead a link from GSL to the clock and flowering. We wish we could provide more mechanistic insight into these non-traditional roles of GSLs but these experiments are still underway. We have included a new section showing that these pleiotropic effects on flowering and indole glucosinolate are not correlated with fitness showing that the GSL to fitness link is not an artifact of some secondary indirect random pleiotropy as best we can measure.

*8) Several points weaken the proposed role of herbivory in shaping the observed patterns in GSL polymorphisms*:

We have largely removed the herbivory arguments because the split plot analysis removed any significance associated with treatment meaning that we don’t have the capacity to support this argument.

*8A) In*
Figure 6
*we see that the ranking of herbivore damage (without insecticide) decreases from UWY2012 > UCD2012 > UWY2011. However, the correlations in*
*Figure 12*
*show the opposite pattern: UWY2012 < UCD2012 < UWY2011. This seems incompatible with the conclusion that the herbivory differences in the field reflect natural selection by herbivores in Eurasian* Arabidopsis *over thousands of generations*.

We have removed the ANCOVA as requested as well as the corresponding figure.

*8B) Similarly, the authors note that* “*the positive correlation between observed genotype frequency and fitness disappeared in the high herbivory WY2012 field trial.*” *This suggests that these results may not be due to herbivory*.

We have removed the ANCOVA as requested as well as the corresponding figure.

*8C) To test for herbivory-mediated effects, one could ask whether the ANCOVA is significant if only the no-pesticide treatment is analyzed. Or, what happens to the patterns in*
Figure 8
*if the change in fitness between insecticide treatments is used as the response variable for the ANCOVA*?

We have removed the ANCOVA as requested by the reviewers.